# Waypoint Transformer: Reinforcement Learning via Supervised Learning with Intermediate Targets

**Anirudhan Badrinath**     **Yannis Flet-Berliac**     **Allen Nie**     **Emma Brunskill**
Department of Computer Science
Stanford University
{abadrina, yfletberliac, anie, ebrun}@cs.stanford.edu

## Abstract

Despite the recent advancements in offline reinforcement learning via supervised learning (RvS) and the success of the decision transformer (DT) architecture in various domains, DTs have fallen short in several challenging benchmarks. The root cause of this underperformance lies in their inability to seamlessly connect segments of suboptimal trajectories. To overcome this limitation, we present a novel approach to enhance RvS methods by integrating intermediate targets. We introduce the Waypoint Transformer (WT), using an architecture that builds upon the DT framework and conditioned on automatically-generated waypoints. The results show a significant increase in the final return compared to existing RvS methods, with performance on par or greater than existing popular temporal difference learning-based methods. Additionally, the performance and stability improvements are largest in the most challenging environments and data configurations, including AntMaze Large Play/Diverse and Kitchen Mixed/Partial.

## 1   Introduction

Traditionally, offline reinforcement learning (RL) methods that compete with state-of-the-art (SOTA) algorithms have relied on objectives encouraging pessimism in combination with value-based methods. Notable examples of this approach include Batch Conservative Q-Learning (BCQ), Conservative Q-Learning (CQL), and Pessimistic Q-Learning (PQL) [Fujimoto et al., 2019, Kumar et al., 2020, Liu et al., 2020]. However, these methods can be challenging to train and often require intricate hyperparameter tuning and various tricks to ensure stability and optimal performance across tasks.

Reinforcement learning via supervised learning (RvS) has emerged as a simpler alternative to traditional offline RL methods [Emmons et al., 2021]. RvS approaches are based on behavioral cloning (BC), either conditional or non-conditional, to train a policy. Importantly, these methods eliminate the need for any temporal-difference (TD) learning, such as fitted value or action-value functions. This results in a simpler algorithmic framework based on supervised learning, allowing for progress in offline RL to build upon work in supervised learning. There are several successful applications of RvS methods, including methods conditioned on goals and returns [Kumar et al., 2019, Janner et al., 2021, Ding et al., 2019, Chen et al., 2021, Emmons et al., 2021].

However, RvS methods have typically struggled in tasks where seamlessly connecting (or "stitching") appropriate segments of suboptimal training trajectories is critical for success [Kumar et al., 2022]. For example, when tasked with reaching specific locations in the AntMaze maze navigation environment or completing a series of tasks in the FrankaKitchen environment, RvS methods typically perform significantly worse than TD learning methods such as Implicit Q-Learning [Fu et al., 2020, Kostrikov et al., 2021].

37th Conference on Neural Information Processing Systems (NeurIPS 2023).

In this study, we leverage the transformer architecture [Vaswani et al., 2017] to construct an RvS method. As introduced by Chen et al. [2021], the decision transformer (DT) can perform conditional behavioral cloning in the context of offline RL. However, similar to other RvS methods, DT proves inferior in performance across popular Gym-MuJoCo benchmarks compared to other value-based offline RL methods, with a 15% relative reduction in average return and lowered stability (Table 1).

To tackle these limitations of existing RvS methods, we introduce a waypoint generation technique that produces intermediate goals and more stable, proxy rewards, which serve as guidance to steer a policy to desirable outcomes. By conditioning a transformer-based RvS method on these generated targets, we obtain a trained policy that learns to follow them, leading to improved performance and stability compared to prior offline RL methods. The highlights of our proposed approach are as follows:

- We propose a novel RvS method, Waypoint Transformer, using waypoint generation networks and establish new state-of-the-art performance, in challenging tasks such as AntMaze Large and Kitchen Partial/Mixed [Fu et al., 2020] (Table 1). On tasks from Gym-MuJoCo, our method rivals the performance of TD learning-based methods such as Implicit Q-Learning and Conservative Q-Learning [Kostrikov et al., 2021, Kumar et al., 2020], and improve over existing RvS methods.

- We motivate the benefit of conditioning RvS on intermediate targets using a chain-MDP example and an empirical analysis of maze navigation tasks. By providing such additional guidance on suboptimal datasets, we show that a policy optimized with a behavioral cloning objective chooses more optimal actions compared to conditioning on fixed targets (as in Chen et al. [2021], Emmons et al. [2021]), facilitating improved stitching capability.

- Our work also provides practical insights for improving RvS, such as significantly reducing training time, solving the hyperparameter tuning challenge in RvS posed by Emmons et al. [2021], and notably improved stability in performance across runs.

## 2   Related Work

Many recent offline RL methods have used fitted value or action-value functions [Liu et al., 2020, Fujimoto et al., 2019, Kostrikov et al., 2021, Kumar et al., 2020, Kidambi et al., 2020, Lyu et al., 2022] or model-based approaches leveraging estimation of dynamics [Kidambi et al., 2020, Yu et al., 2020, Argenson and Dulac-Arnold, 2020, Shen et al., 2021, Rigter et al., 2022, Zhan et al., 2021].

RvS, as introduced in Emmons et al. [2021], avoids fitting value functions and instead leverages behavioral cloning. In many RvS-style methods, the conditioning variable for the policy is based on the return [Kumar et al., 2019, Srivastava et al., 2019, Schmidhuber, 2019, Chen et al., 2021], but other methods use goal-conditioning [Nair et al., 2018, Emmons et al., 2021, Ding et al., 2019, Ghosh et al., 2019] or leverage inverse RL [Eysenbach et al., 2020]. Recent work by Brandfonbrener et al. [2022] has explored the limitations of reward conditioning in RvS. In this study, we consider both reward and goal-conditioning.

Transformers have demonstrated the ability to generalize to a vast array of tasks, such as language modeling, image generation, and representation learning [Vaswani et al., 2017, Devlin et al., 2018, He et al., 2022, Parmar et al., 2018]. In the context of offline RL, decision transformers (DT) leverage a causal transformer architecture to fit a reward-conditioned policy [Chen et al., 2021]. Similarly, [Janner et al., 2021] frame offline RL as a sequence modeling problem and introduce the Trajectory Transformer, a model-based offline RL approach that uses the transformer architecture.

Algorithms building upon the DT, such as online DT [Zheng et al., 2022], prompt DT [Xu et al., 2022] and Q-Learning DT [Yamagata et al., 2022], have extended the scope of DT's usage. Furuta et al. [2021] introduce a framework for hindsight information matching algorithms to unify several hindsight-based algorithms, such as Hindsight Experience Replay [Andrychowicz et al., 2017], DT, TT, and our proposed method.

Some critical issues with DT unresolved by existing work are (a) its instability (i.e., large variability across initialization seeds) for some tasks in the offline setting (Table 1) and (b) its relatively poor performance on some tasks due to an inability to stitch segments of suboptimal trajectories [Kumar et al., 2022] – in such settings, RvS methods are outperformed by value-based methods such as

Implicit Q-Learning (IQL) [Kostrikov et al., 2021] and Conservative Q-Learning (CQL) [Kumar et al., 2020]. We address both these concerns with our proposed approach, demonstrating notably improved performance and reduced variance across seeds for tasks compared to DT and prior RvS methods (Table 1).

One of the areas of further research in RvS, per Emmons et al. [2021], is to address the complex and unreliable process of tuning hyperparameters, as studied in Zhang and Jiang [2021] and Nie et al. [2022]. We demonstrate that our method displays low sensitivity to changes in hyperparameters compared to Emmons et al. [2021] (Table 2). Further, all experiments involving our proposed method use the same set of hyperparameters in achieving SOTA performance across many tasks (Table 1).

## 3  Preliminaries

We assume that there exists an agent interacting with a Markov decision process (MDP) with states $s_t \in \mathcal{S}$ and actions $a_t \in \mathcal{A}$ with unknown transition dynamics $p(s_{t+1} \mid s_t, a_t)$ and initial state distribution $p(s_0)$. The agent chooses an action sampled from a transformer policy $a_t \sim \pi_\theta(a_t \mid s_{t-k..t}, \Phi_{t-k..t})$, parameterized by $\theta$ and conditioned on the known history of states $s_{t-k..t}$ and a conditioning variable $\Phi_{t-k..t}$. Compared to the standard RL framework, where the policy is modeled by $\pi(a_t \mid s_t)$, we leverage a policy that considers past states within a fixed context window $k$.

The conditioning variable $\Phi_t$ is a specification of a goal or reward based on a target outcome $\omega$. At training time, $\omega$ is sampled from the data, as presented in Emmons et al. [2021]. At test time, we assume that we can either generate or are provided global goal information $\omega$ for goal-conditioned tasks. For reward-conditioned tasks, we specify a target return $\omega$, following Chen et al. [2021].

To train our RvS-based algorithm on an offline dataset $\mathcal{D}$ consisting of trajectories $\tau$ with conditioning variable $\Phi_t$, we compute the output of an autoregressive transformer model based on past and current states and conditioning variable provided in each trajectory. Using a negative log-likelihood loss, we use gradient descent to update the policy $\pi_\theta$. This procedure is summarized in Algorithm 1.

---

**Algorithm 1** Training algorithm for transformer-based policy trained on offline dataset $\mathcal{D}$.

---

**Input:** Training dataset $\mathcal{D} = \{\tau_1, \tau_2, \tau_3, ..., \tau_n\}$ of training trajectories.
**for** each $\tau = (s_0, a_0, \Phi_0, s_1, a_1, \Phi_1, ...)$ in $\mathcal{D}$ **do**
    Compute $\pi_\theta(a_t \mid s_{t-k..t}, \Phi_{t-k..t})$ for all $t$
    Calculate $L_\theta(\tau) = -\sum_t \log \pi_\theta(a_t \mid s_{t-k..t}, \Phi_{t-k..t})$
    Backpropagate gradients w.r.t. $L_\theta(\tau)$ to update model parameters
**end for**

---

## 4  Waypoint Generation

In this section, we propose using intermediate targets (or waypoints) as conditioning variables as an alternative to fixed targets, proposed in Emmons et al. [2021]. Below, we motivate the necessity for waypoints in RvS and present a practical technique to generate waypoints that can be used for goal-conditioned (Section 4.2) and reward-conditioned tasks (Section 4.3) respectively.

### 4.1  Illustrative Example

To motivate the benefits of using waypoints, we consider an infinite-horizon, deterministic MDP with $H+1$ states and two possible actions at non-terminal states. A graphical representation of the MDP is shown in Figure 1. For this scenario, we consider the goal-conditioned setting where the target goal state during train and test time is $\omega = s^{(H)}$, and the episode terminates once we reach $\omega$.

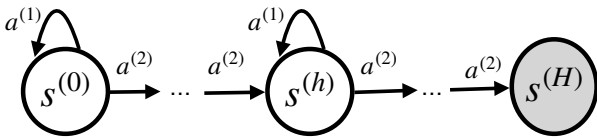

Figure 1: Chain MDP to motivate the benefit of intermediate goals for conditional BC-based policy training.

In offline RL, the data is often suboptimal for achieving the desired goal during testing. In this example, suppose we have access to a dataset $\mathcal{D}$ that contains an infinite number of trajectories

collected by a random behavioral policy $\pi_b$ where $\pi_b(a_t = a^{(1)} \mid s_t) = \lambda > 0$ for all $s_t$. Clearly, $\pi_b$ is suboptimal with respect to reaching $\omega$ in the least number of timesteps; in expectation, it takes $\frac{H}{1-\lambda}$ timesteps to reach $s^{(H)}$ instead of $H$ (optimal) since the agent "stalls" at the current state with probability $\lambda$ and moves to the next state with probability $1 - \lambda$.

Consider a global goal-conditioned policy $\pi_G(a_t \mid s_t, \omega)$ that is optimized using a behavioral cloning objective on $\mathcal{D}$. Clearly, the optimal policy $\pi_G^*(a_t \mid s_t, \omega) = \pi_b(a_t \mid s_t) \; \forall s_t$ since $\omega = s^{(H)}$ is a constant. Hence, the global goal-conditioned policy $\pi_G^*$ is as suboptimal as the behavioral policy $\pi_b$.

Instead, suppose that we condition a policy $\pi_W(a_t \mid s_t, \Phi_t)$ on an intermediate goal state $\Phi_t = s_{t+K}$ for some chosen $K < \frac{1}{1-\lambda}$ (expected timesteps before $\pi_b$ executes $a_2$), optimized using a behavioral cloning objective on $\mathcal{D}$. For simplicity, suppose our target intermediate goal state $\Phi_t$ for some current state $s_t = s^{(h)}$ is simply the next state $\Phi_t = s^{(h+1)}$. Based on data $\mathcal{D}$ from $\pi_b$, the probability of taking action $a^{(2)}$ conditioned on the chosen $\Phi_t$ and $s_t$ is estimated as:

$$\mathrm{Pr}_{\pi_b}[a_t = a^{(2)} \mid s_t = s^{(h)}, s_{t+K} = s^{(h+1)}] = \frac{\mathrm{Pr}_{\pi_b}[a_t = a^{(2)}, s_{t+K} = s^{(h+1)} \mid s_t = s^{(h)}]}{\mathrm{Pr}_{\pi_b}[s_{t+K} = s^{(h+1)} \mid s_t = s^{(h)}]}$$

$$= \frac{(1-\lambda)\lambda^{K-1}}{\binom{K}{1}[(1-\lambda)\lambda^{K-1}]} = \frac{1}{\binom{K}{1}} = \frac{1}{K}.$$

Hence, for the optimal intermediate goal-conditioned policy $\pi_W^*$ trained on $\mathcal{D}$, the probability of choosing the optimal action $a^{(2)}$ is:

$$\pi_W^*(a_t = a^{(2)} \mid s_t = s^{(h)}, \Phi_t = s^{(h+1)}) = \frac{1}{K}.$$

Since $\pi_G^*(a_t = a^{(2)} \mid s_t = s^{(h)}, \omega) = 1 - \lambda$ and we choose $K$ such that $\frac{1}{K} > 1 - \lambda$, we conclude:

$$\pi_W^*(a_t = a^{(2)} \mid s_t, \Phi_t) > \pi_G^*(a_t = a^{(2)} \mid s_t, \omega).$$

The complete derivation is presented in Appendix A. Based on this example, conditioning the actions on reaching a desirable intermediate state is more likely to result in taking the optimal action compared to a global goal-conditioned policy. Effectively, the conditioning acts as a "guide" for the policy, directing it toward desirable intermediate targets in order to reach the global goal.

## 4.2 Intermediate Goal Generation for Spatial Compositionality

In this section, we address RvS's inability to "stitch" subsequences of suboptimal trajectories in order to achieve optimal behaviour, based on analyses in Kumar et al. [2022]. In that pursuit, we introduce a technique to generate effective intermediate targets to better facilitate stitching and to guide the policy towards desirable outcomes, focusing primarily on goal-conditioned tasks.

Critically, the ability to stitch requires considering experiences that are more relevant to achieving appropriate short-term goals before reaching the global goal. To illustrate this, we show a maze navigation task from the AntMaze Large environment in Figure 2, where the evaluation objective is to reach a target location from the start location [Fu et al., 2020].

Analyzing the training trajectories that pass through either the start location (blue) or target location (red), less than 5% of trajectories extend beyond the stitching region into the other region, i.e., the target or start regions respectively. Since trajectories seldom pass through both the start and target regions, the policy must "stitch" together subsequences from the blue and red trajectories within the stitching region, where the trajectories overlap the most. By providing intermediate targets within this region, rather than conditioning solely on the global goal, we can guide the policy to connect the relevant subsequences needed to reach the target effectively.

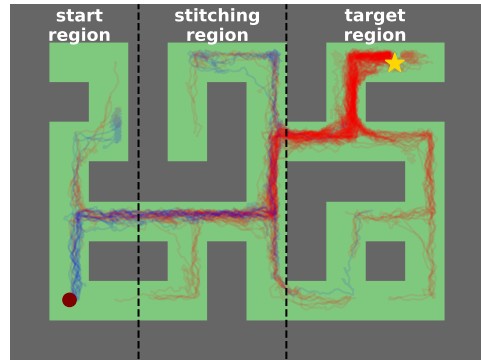

Figure 2: `antmaze-large-play-v2` task to navigate from the start location (circle) to the target location (star). Blue and red colored lines are training trajectories passing through the start or end locations respectively.

To obtain effective intermediate targets, we propose the goal waypoint network, explicitly designed to generate short-term, intermediate goals. Similar to the illustrative example in Section 4.1, the purpose of these intermediate targets is to guide the policy network $\pi_\theta$ towards states that lead to the desired global goal by facilitating stitching of relevant subsequences.

To that end, we represent the goal waypoint network $W_\phi$, parameterized by $\phi$, as a neural network that makes approximate $K$-step predictions of future observations conditioned on the current state, $s_t$, and the target goal, $\omega$. Formally, we attempt to minimize the objective in Equation 1 across the same offline dataset $\mathcal{D}$, where $L_\phi$ is a mean-squared error (MSE) loss for continuous state spaces:

$$\arg\min_\phi \sum_{\tau \in \mathcal{D}} L_\phi(W_\phi(s_t, \omega), s_{t+K}). \tag{1}$$

While our approach to intermediate target generation seems simple in relation to the complex problem of modeling both the behavioral policy and transition dynamics, our goal is to provide approximate short-term goals to facilitate the downstream task of reaching the global goal $\omega$, rather than achieving perfect predictions of future states under the behavioral policy.

### 4.3 Proxy Reward Generation for Bias-Variance Reduction

In this section, we address the high bias and variance of conditioning variables used by prior RvS methods in reward-conditioned tasks, such as Emmons et al. [2021] and Chen et al. [2021]. Analogously to Section 4.2 (i.e., for goal-conditioned tasks), we propose a technique to generate intermediate reward targets for reward-conditioned tasks to mitigate these issues.

Existing methods rely on either an initial cumulative reward-to-go (desired return) or an average reward-to-go target, denoted as $\omega$. Importantly, the former is updated using rewards obtained during the rollout, while the latter remains constant over time [Emmons et al., 2021, Chen et al., 2021]. However, using these conditioning variables during evaluation gives rise to two main issues: (a) the Monte Carlo estimate of return used to compute the cumulative reward-to-go exhibits high variance and (b) the constant average reward-to-go target introduces high bias over time. Based on our analyses of the bias and variance of these approaches in Appendix C, we observe that these issues contribute to decreased performance and stability across runs when evaluating RvS methods.

Although a potential approach to mitigate these issues is to leverage TD learning, such as the Q-Learning Transformer [Yamagata et al., 2022], we restrict our work to RvS methods utilizing behavioral cloning objectives due to the inherent complexity of training value-based methods. To address the aforementioned concerns, we introduce a reward waypoint network denoted as $W_\phi$, parameterized by $\phi$. This network predicts the average and cumulative reward-to-go (`ARTG`, `CRTG`) conditioned on the return, $\omega$, and current state, $s_t$, using offline data $\mathcal{D}$. To optimize this network, we minimize the objective shown in Equation 2 using an MSE loss:

$$\arg\min_\phi \sum_{\tau \in \mathcal{D}} \left( \left[ \frac{1}{T-t} \sum_{t'=t}^{T} \gamma^t r_t \quad \sum_{t'=t}^{T} \gamma^t r_t \right]^\top - W_\phi(s_t, \omega) \right)^2. \tag{2}$$

By modeling both `ARTG` and `CRTG`, we address the high bias of a constant `ARTG` target and reduce the variance associated with Monte Carlo estimates for `CRTG`. The construction of the reward waypoint network is similar in motivation and the prediction task to a baseline network, used to mitigate high variance in methods like REINFORCE [Sutton and Barto, 1999]. However, the distinguishing feature of our reward waypoint network lies in its conditioning on the return, which allows for favorable performance even on suboptimal offline datasets.

## 5 Waypoint Transformer

We propose the waypoint transformer (WT), a transformer-based offline RL method that leverages the proposed waypoint network $W_\phi$ and a GPT-2 architecture based on multi-head attention [Radford et al., 2019]. The WT policy $\pi_\theta$ is conditioned on past states $s_{t-k..t}$ and waypoints (either generated goals or rewards) $\Phi_{t-k..t} = W_\phi(s_{t-k..t}, \omega)$ with a context window of size $k$, as shown in Figure 3.

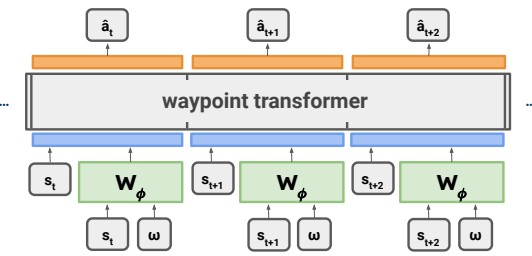

Figure 3: Waypoint Transformer architecture, where $\Phi_t = W_\phi(s_t, \omega)$ represents the output of the goal or reward waypoint network.

Table 1: Normalized scores and training time per task on Gym-MuJoCo, AntMaze, and Kitchen tasks, where **bold** highlighting indicates SOTA performance, defined by a method's average performance being contained within the interval of the method with highest average.

| Environment | TD3 + BC | Onestep RL | CQL | IQL | BC | 10% BC | RvS-R/G | DT | QDT | WT (Ours) |
|---|---|---|---|---|---|---|---|---|---|---|
| halfcheetah-medium-v2 | **48.3 ± 0.3** | **48.4 ± 0.1** | 44.0 ± 5.4 | 47.4 ± 0.2 | 42.6 | 42.5 | 41.6 ± 0.3 | 42.4 ± 0.2 | 42.3 ± 0.4 | 43.0 ± 0.2 |
| hopper-medium-v2 | 59.3 ± 4.2 | 59.6 ± 2.5 | 58.5 ± 2.1 | **66.2 ± 5.7** | 52.9 | 56.9 | 60.2 ± 3.0 | 63.5 ± 5.2 | **66.5 ± 6.3** | 63.1 ± 1.4 |
| walker2d-medium-v2 | 83.7 ± 2.1 | **81.8 ± 2.2** | 72.5 ± 0.8 | 78.3 ± 8.7 | 75.3 | 75.0 | 71.7 ± 1.8 | 69.2 ± 4.9 | 67.1 ± 3.2 | 74.8 ± 1.0 |
| halfcheetah-medium-replay-v2 | 44.6 ± 0.5 | 38.1 ± 1.3 | **45.5 ± 0.5** | 44.2 ± 1.2 | 36.6 | 40.6 | 38.0 ± 0.7 | 35.4 ± 1.6 | 35.6 ± 0.5 | 39.7 ± 0.3 |
| hopper-medium-replay-v2 | 60.9 ± 18.8 | **97.5 ± 0.7** | 95.0 ± 6.4 | 94.7 ± 8.6 | 18.1 | 75.9 | 73.5 ± 12.8 | 43.3 ± 23.9 | 52.1 ± 20.1 | 88.9 ± 2.4 |
| walker2d-medium-replay-v2 | **81.8 ± 5.5** | 49.5 ± 12.0 | 77.2 ± 5.5 | 73.8 ± 7.1 | 26.0 | 62.5 | 60.6 ± 6.7 | 58.9 ± 7.1 | 58.2 ± 5.1 | 67.9 ± 3.4 |
| halfcheetah-medium-expert-v2 | 90.7 ± 4.3 | **93.4 ± 1.6** | 91.6 ± 2.8 | 86.7 ± 5.3 | 55.2 | 92.9 | 92.2 ± 1.2 | 84.9 ± 1.6 | - | **93.2 ± 0.5** |
| hopper-medium-expert-v2 | 98.0 ± 9.4 | 103.3 ± 1.9 | 105.4 ± 6.8 | 91.5 ± 14.3 | 52.5 | **110.9** | 101.7 ± 16.5 | 100.6 ± 8.3 | - | **110.9 ± 0.6** |
| walker2d-medium-expert-v2 | 110.1 ± 0.5 | **113.0 ± 0.4** | 108.8 ± 0.7 | 109.6 ± 1.0 | 107.5 | 109.0 | 106.0 ± 0.9 | 89.6 ± 38.4 | - | 109.6 ± 1.0 |
| gym-avg-v2 | **75.3 ± 4.9** | **76.1 ± 2.5** | 77.6 ± 3.4 | 76.9 ± 5.8 | 51.9 | 74.0 | 71.7 ± 4.9 | 65.3 ± 10.1 | - | **76.8 ± 1.2** |
| antmaze-umaze-v2 | 78.6 | 64.3 | 74.0 | **87.5 ± 2.6** | 54.6 | 62.8 | 65.4 ± 4.9 | 53.6 ± 7.3 | - | 64.9 ± 6.1 |
| antmaze-umaze-diverse-v2 | 71.4 | 60.7 | **84.0** | 62.2 ± 13.8 | 45.6 | 50.2 | 60.9 ± 2.5 | 42.2 ± 5.4 | - | 71.5 ± 7.6 |
| antmaze-medium-play-v2 | 10.6 | 0.3 | 61.2 | **71.2 ± 7.3** | 0.0 | 5.4 | 58.1 ± 12.7 | 0.0 ± 0.0 | - | 62.8 ± 5.8 |
| antmaze-medium-diverse-v2 | 3.0 | 0.0 | 53.7 | **70.0 ± 10.9** | 0.0 | 9.8 | 67.3 ± 8.0 | 0.0 ± 0.0 | - | 66.7 ± 3.9 |
| antmaze-large-play-v2 | 0.2 | 0.0 | 15.8 | 39.6 ± 5.8 | 0.0 | 0.0 | 32.4 ± 10.5 | 0.0 ± 0.0 | - | 72.5 ± 2.8 |
| antmaze-large-diverse-v2 | 0.0 | 0.0 | 14.9 | 47.5 ± 9.5 | 0.0 | 6.0 | 36.9 ± 4.8 | 0.0 ± 0.0 | - | 72.0 ± 3.4 |
| antmaze-avg-v2 | 27.3 | 20.9 | 50.6 | 63.0 ± 8.3 | 16.7 | 22.5 | 53.5 ± 7.2 | 16.0 ± 2.1 | - | **68.4 ± 4.9** |
| kitchen-complete-v0 | - | - | 43.8 | **62.5** | 65.0 | 4.0 | 50.2 ± 3.6 | 46.5 ± 3.0 | - | 49.2 ± 4.6 |
| kitchen-partial-v0 | - | - | 49.8 | 46.3 | 38.0 | **66.0** | 51.4 ± 2.6 | 31.4 ± 19.5 | - | 63.8 ± 3.5 |
| kitchen-mixed-v0 | - | - | 51.0 | 51.0 | 51.5 | 40.0 | 60.3 ± 9.4 | 25.8 ± 5.0 | - | 70.9 ± 2.1 |
| kitchen-avg-v0 | - | - | 48.2 | 53.3 ± 7.5 | 51.5 | 36.7 | 54.0 ± 5.2 | 34.6 ± 9.2 | - | 61.3 ± 3.4 |
| average | - | - | 63.7 | 68.3 ± 6.9 | 40.1 | 50.6 | 62.7 ± 5.7 | 43.8 ± 7.3 | - | 71.4 ± 2.8 |
| training time (min) | 20 | 20 | 80 | 20 | 10 | 10 | 80 | 150 | - | 20 |

For goal-conditioned and reward-conditioned tasks, we train the goal and reward waypoint network $W_\phi$ respectively on offline dataset $\mathcal{D}$ independently of the policy. To train the WT policy, we use Algorithm 1 to iteratively optimize its parameters $\theta$. During this process, the trained weights $\phi$ of $W_\phi$ are frozen to ensure the interpretability of the waypoint network's generated goal and reward waypoints. To further simplify the design and improve computational efficiency, the WT is not conditioned on past actions $a_{t-k..t}$ (i.e., unlike the DT) and we concatenate $\Phi_t$ with $s_t$ to produce one token per timestep $t$ instead of multiple tokens as proposed in Chen et al. [2021].

# 6 Experiments

We present a series of evaluations of WT across tasks involving reward and goal-conditioning, with comparisons to prior offline RL methods. For this, we leverage D4RL, an open-source benchmark for offline RL, consisting of varying datasets for tasks from Gym-MuJoCo, AntMaze, and FrankaKitchen [Fu et al., 2020].

Tasks in the AntMaze and FrankaKitchen environments have presented a challenge for offline RL methods as they contain little to no optimal trajectories and perform critical evaluations of a model's stitching ability [Fu et al., 2020]. Specifically, in FrankaKitchen, the aim is to interact with a set of kitchen items to reach a target configuration, but the `partial` and `mixed` offline datasets consist of suboptimal, undirected data, where the demonstrations are unrelated to the target configuration. Similarly, AntMaze is a maze navigation environment with sparse rewards, where the `play` and `diverse` datasets contain target locations unaligned with the evaluation task. For our experiments on these environments, we use goal-conditioning on the target goal state (i.e., $\omega = s_{\text{target}}$), constructing intermediate targets with the goal waypoint network.

Gym-MuJoCo serves as a popular benchmark for prior work in offline RL, consisting of environments such as `Walker2D`, `HalfCheetah`, and `Hopper`. Importantly, we evaluate across offline datasets with varying degrees of optimality by considering the `medium`, `medium-replay`, and `medium-expert` datasets [Fu et al., 2020]. For these tasks, we use reward-conditioning given a target return, constructing intermediate reward targets using the reward waypoint network; as noted in Emmons et al. [2021], we find that the notion of goals in undirected locomotion tasks is ill-defined.

Across all environments and tasks, we use the same set of hyperparameters, as reported in Appendix B. To measure the stability (i.e., variability) in our method across random initializations, we run each experiment across 5 seeds and report the mean and standard deviation.

## 6.1 Comparing WT with Prior Methods

To evaluate the performance of WT, we perform comparisons with prior offline RL methods, including conditional BC methods such as DT and RvS-R/G; value-based methods such as Onestep RL

[Brandfonbrener et al., 2021], TD3 + BC [Fujimoto and Gu, 2021], CQL, and IQL; and standard BC baselines. For all methods except DT, we use reported results from Emmons et al. [2021] and Kostrikov et al. [2021]. We evaluate DT using the official implementation provided by Chen et al. [2021] across 5 random initializations, though we are unable to reproduce some of their results.

Table 1 shows the results of our comparisons to prior methods. Aggregated across all tasks, WT (71.4 $\pm$ 2.8) improves upon the next best method, IQL (68.3 $\pm$ 6.9), with respect to average normalized score and achieves equal-best runtime. In terms of variability across seeds, there is a notable reduction compared to IQL and most other methods.

In the most challenging tasks requiring stitching, our method demonstrates performance far exceeding the next best method, IQL. On the AntMaze Large datasets, WT demonstrates a substantial relative percentage improvement of 83.1% (`play`) and 51.6% (`diverse`). On Kitchen Partial and Mixed, the improvement is 37.8% and 39.0% respectively. WT's standard deviation across seeds is reduced by a factor of more than 2x compared to IQL for these tasks.

Similarly, on reward-conditioned tasks with large performance gaps between BC and value-based methods such as `hopper-medium-replay-v2`, WT demonstrates increased average performance by 105.3% compared to DT and 21.0% compared to RvS-R, with standard deviation reduced by a factor of 10.0x and 5.3x respectively.

## 6.2 Utility of Waypoint Networks

To analyze the utility and behavior of waypoint networks, we qualitatively evaluate an agent's performance across rollouts of trained transformer policies on `antmaze-large-play-v2`. For this analysis, we consider a WT policy (using a goal waypoint network with $K = 30$) and a global goal-conditioned transformer policy (i.e., no intermediate goals). Across both models, the architecture and hyperparameters for training are identical.

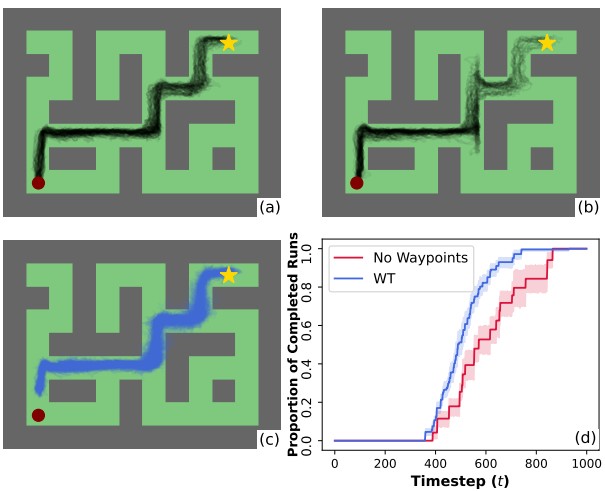

The ant's locations across 100 rollouts of a WT policy (Figure 4a) and a global goal-conditioned transformer policy (Figure 4b) demonstrate that WT shows notably higher ability and consistency in reaching the goal location. Specifically, without intermediate goals, the ant occasionally turns in the wrong direction and demonstrates a lesser ability to successfully complete a turn based on the reduction of density at each turn (Figure 4b). Consequently, the

Figure 4: Shows the ant's location across 100 rollouts of **(a)** a WT policy and **(b)** a global goal-conditioned transformer policy; **(c)** generated intermediate goals by the waypoint network $W_\phi$, **(d)** the proportion of all successful runs completed by timestep $t$.

WT achieves more than twice the evaluation return (72.5 $\pm$ 2.8) compared to the global goal-conditioned policy (33.0 $\pm$ 10.3) and completes the task more quickly on average (Figure 4d).

Based on Figure 4c, we observe that the goal waypoint network provides goals that correspond to the paths traversed in Figure 4a for the WT policy. This shows that the waypoint network successfully guides the model toward the target location, addressing the stitching problem proposed in Figure 2. While the global goal-conditioned policy is successful in passing beyond the stitching region into the target region in only 45% of the rollouts, accounting for 82% of its failures to reach the target, WT is successful in this respect for 87% of rollouts.

## 6.3 Ablation Studies

**Goal-Conditioned Tasks** On goal-conditioned tasks, we examine the behavior of the goal waypoint network as it relates to the performance of the policy at test time by ablating aspects of its configuration

and training. For this analysis, we consider `antmaze-large-play-v2`, a challenging task that critically evaluates the stitching capability of offline RL techniques.

To understand the effect of the configuration of the goal waypoint network on test performance, we ablate two variables relevant to generating effective intermediate goals: the temporal proximity of intermediate goals ($K$) and the validation loss of the goal waypoint network.

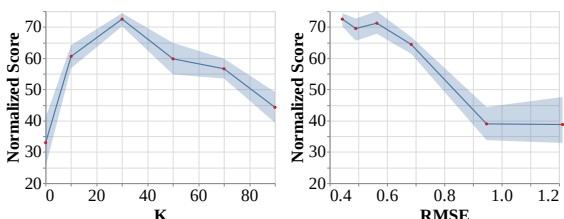

The normalized score attained by the agent is shown as a function of $K$ and the validation loss of the goal waypoint network in Figure 5. For this environment and dataset, an ideal choice for $K$ is around 30 timesteps. For all nonzero $K$, the performance is reduced at a reasonably consistent rate on either side of $K = 30$. Importantly, when $K = 0$ (i.e., no intermediate goals), there is a notable reduction in performance compared to all other choices of $K$; compared to the optimal $K = 30$, the score is reduced by a factor of 2.2x.

Figure 5: Normalized score attained by WT on `antmaze-large-play-v2` based on varying **left**: the temporal proximity of generated goals, $K$, and **right**: goal waypoint network RMSE on a held-out dataset.

In Figure 5 (right), the normalized score shows the negligible change for values of held-out RMSE between 0.4 and 0.6, corresponding to at least 1,000 gradient steps or roughly 30 sec of training, with a sharper decrease henceforth. As the RMSE increases to over 1, we observe a relative plateau in performance near an average normalized score of 35-45, roughly corresponding to performance without using a waypoint network (i.e., $K = 0$ in Figure 5 (left)).

Additionally, we perform comparisons between the goal waypoint network and manually constructed waypoints as intermediate targets for WT, for which the methodology and results are shown in Appendix D. Based on that analysis, we show that manual waypoints statistically significantly improve upon no waypoints ($44.5 \pm 2.8$ vs. $33.0 \pm 10.3$), but they remain significantly worse than generated waypoints.

**Reward-Conditioned Tasks** On reward-conditioned tasks, we ablate the choice of different reward-conditioning techniques. Specifically, we examine the performance of WT and variance of the reward waypoint network in comparison to `CRTG` updated using the rewards obtained during rollouts and a static `ARTG` (i.e., as done in Chen et al. [2021] and Emmons et al. [2021]). We consider the `hopper-medium-replay-v2` task for this analysis as there is (a) a large performance gap between RvS and value-based methods, and (b) high instability across seeds for RvS methods (e.g., DT) as shown in Table 1. For all examined reward-conditioned techniques, the transformer architecture and training procedure are identical, and the target (normalized) return is 95, corresponding to SOTA performance.

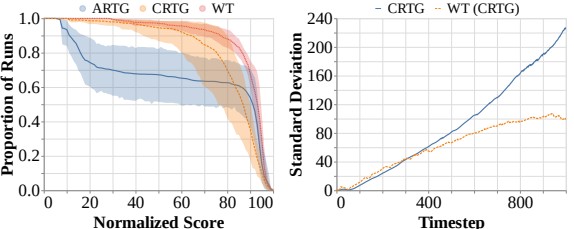

Figure 6: Comparison of different reward-conditioning methods on `hopper-medium-replay-v2`. **Left**: Performance profiles for transformers using `ARTG`, `CRTG`, and WT across 5 random seeds. **Right**: Standard deviation in `CRTG` inputted to the model when updated with attained rewards ($\text{CRTG}_t = \omega - \sum_t \gamma^t r_t$) and using predictions from the reward waypoint network ($W_\phi(s_t, \omega)_2$) when average return is approximately held constant.

To examine the distribution of normalized scores across different seeds produced by each of the described reward-conditioning techniques, we construct performance profiles, displaying the proportion of runs greater than a certain normalized score [Agarwal et al., 2021]. As shown in Figure 6 (left), WT demonstrates increased performance and stability across random initializations compared to the remaining reward-conditioning techniques.

Additionally, we perform an analysis to determine whether using a reward waypoint network to predict the `CRTG` as opposed to updating the `CRTG` using attained rewards as in Chen et al. [2021] affects the variability of the conditioning variable passed to the policy network (i.e., not of the

performance as that is examined in Figure 6). Importantly, to account for performance differences between the policies trained with either method that may influence the variability of the attained CRTG, we sample a subset of runs for both methods such that the average performance is constant. Based on Figure 6 (right), it is clear that as a function of the timestep, when accounting for difference in average performance, the standard deviation in the CRTG predicted by WT grows at a slower rate compared to updating CRTG with attained rewards.

**Transformer Configuration**    Based on the work in Emmons et al. [2021], we balance between expressiveness and regularization to maximize policy performance. We ablate the probability of node dropout $p_{\text{drop}}$ and the number of transformer layers $L$. To further examine this balance, we experiment with conditioning on past actions $a_{t-k..t-1}$, similarly to the DT, to characterize its impact on performance and computational efficiency. In this section, we consider antmaze-large-play-v2, hopper-medium-replay-v2 and kitchen-mixed-v0, one task from each category of environments.

Based on Table 2, we observe that the sensitivity to the various ablated hyperparameters is relatively low in terms of performance, and removing action conditioning results in reduced training time and increased performance, perhaps due to reduced distribution shift at evaluation. In context of prior RvS work where dropout ($p_{\text{drop}} = 0.1$) decreased performance compared to no dropout by 1.5-3x on AntMaze, the largest decrease in average performance on WT is only by a factor of 1.1x [Emmons et al., 2021].

Table 2: Ablation of transformer configuration showing normalized score on MuJoCo (v2), AntMaze (v2) and Kitchen (v0), including dropout ($p_{\text{drop}}$), transformer layers ($L$), and action conditioning ($a_t$), where bolded hyperparameters (e.g., 0.150) are used for final models and bolded scores are optimal.

| $p_{\text{drop}}$ | hopper-medium-replay | antmaze-large-play | kitchen-mixed | Average |
|---|---|---|---|---|
| 0.000 | $75.5 \pm 8.3$ | $68.3 \pm 5.9$ | $\mathbf{72.9 \pm 0.5}$ | $72.2 \pm 4.9$ |
| 0.075 | $\mathbf{89.8 \pm 2.8}$ | $70.8 \pm 4.5$ | $71.8 \pm 1.2$ | $\mathbf{77.5 \pm 2.8}$ |
| **0.150** | $88.9 \pm 2.4$ | $72.5 \pm 2.8$ | $70.9 \pm 2.1$ | $77.4 \pm 2.4$ |
| 0.225 | $75.7 \pm 9.4$ | $72.2 \pm 2.7$ | $71.2 \pm 1.0$ | $73.0 \pm 4.4$ |
| 0.300 | $74.7 \pm 10.2$ | $73.5 \pm 2.5$ | $69.2 \pm 2.0$ | $72.5 \pm 4.9$ |
| 0.600 | $58.4 \pm 7.5$ | $\mathbf{73.8 \pm 5.2}$ | $66.5 \pm 2.7$ | $66.2 \pm 5.1$ |

| $L$ | hopper-medium-replay | antmaze-large-play | kitchen-mixed | Average |
|---|---|---|---|---|
| 1 | $82.1 \pm 8.8$ | $72.1 \pm 5.7$ | $\mathbf{71.6 \pm 1.6}$ | $75.3 \pm 5.4$ |
| **2** | $88.9 \pm 2.4$ | $\mathbf{72.5 \pm 2.8}$ | $70.9 \pm 2.1$ | $\mathbf{77.4 \pm 2.4}$ |
| 3 | $89.9 \pm 1.6$ | $71.8 \pm 3.0$ | $70.3 \pm 2.1$ | $77.3 \pm 2.2$ |
| 4 | $\mathbf{91.1 \pm 2.8}$ | $65.8 \pm 3.8$ | $69.7 \pm 1.0$ | $75.5 \pm 2.5$ |
| 5 | $88.8 \pm 4.5$ | $66.7 \pm 4.7$ | $70.0 \pm 0.8$ | $75.2 \pm 3.3$ |

| $a_t$ | hopper-medium-replay | antmaze-large-play | kitchen-mixed | Average |
|---|---|---|---|---|
| Yes | $76.9 \pm 9.0$ | $66.5 \pm 5.6$ | $65.2 \pm 2.8$ | $69.5 \pm 5.8$ |
| **No** | $\mathbf{88.9 \pm 2.4}$ | $\mathbf{72.5 \pm 2.8}$ | $\mathbf{70.9 \pm 2.1}$ | $\mathbf{77.4 \pm 2.4}$ |

## 7   Discussion

In this study, we address the issues with existing conditioning techniques used in RvS, such as the "stitching" problem associated with global goals and the high bias and variance of reward-to-go targets, through the automatic generation of intermediate targets. Based on empirical evaluations, we demonstrate significantly improved performance and stability compared to existing RvS methods, often on par with or outperforming TD learning methods. Especially on challenging tasks with suboptimal dataset composition, such as AntMaze Large and Kitchen Partial/Mixed, the guidance provided by the waypoint network through intermediate targets (e.g., as shown in Figure 4) significantly improves upon existing state-of-the-art performance.

We believe that this work can present a pathway forward to developing practical offline RL methods leveraging the simplicity of RvS and exploring more effective conditioning techniques, as formalized by Emmons et al. [2021]. In addition to state-of-the-art performance, we demonstrate several desirable practical qualities of the WT: it is less sensitive to changes in hyperparameters, significantly faster to train than prior RvS work, and more consistent across initialization seeds.

However, despite improvements across challenging tasks, WT's margin of improvement on AntMaze U-Maze and Kitchen Complete (i.e., easier tasks) is lower: its normalized scores are more comparable to DT and other RvS methods. We believe this is likely due to stitching being less necessary in such tasks compared to difficult tasks, rendering the impact of the waypoint network negligible. To further characterize the performance of the waypoint networks and WT on such tasks is an interesting direction for future work. In addition, there are several limitations inherited by the usage of the RvS framework, such as manual tuning of the target return at test time for reward-conditioned tasks using a grid search, issues with stochasticity, and an inability to learn from data with multimodal outcomes.

## 8 Conclusion

We propose a method for reinforcement learning via supervised learning, Waypoint Transformer, conditioned on generated intermediate targets for reward and goal-conditioned tasks. We show that RvS with waypoints significantly surpasses existing RvS methods and achieves on par with or surpasses popular state-of-the-art methods across a wide range of tasks from Gym-MuJoCo, AntMaze, and Kitchen. With improved stability across runs and competitive computational efficiency, we believe that our method advances the performance and applicability of RvS within the context of offline RL.

## Acknowledgments and Disclosure of Funding

This work was supported in part by NSF grant #2112926.

We thank Scott Emmons and Ilya Kostrikov for their discussions on and contributions to providing results for prior offline RL methods.

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

# A Derivation for Illustrative Example

We provide detailed derivations based on the simple deterministic MDP shown in Section 4.1, in the context of an offline dataset $\mathcal{D}$ collected by a random behavioural policy $\pi_b$. We show that minimization of a maximum likelihood objective on $\pi_G$ yields $\pi_b$, the behavioural policy. Note $\pi_G(a_t \mid s_t, \omega) = \pi_G(a_t \mid s_t)$ as $\omega = s^{(H)}$ is a constant (and as a result, $a_t$ is conditionally independent). To obtain the optimal policy $\pi_G^*$, we maximize the following objective:

$$\arg \max_{\pi_G} \mathbb{E}_{(s_t, a_t) \in \mathcal{D}}[\log \pi_G(a_t \mid s_t, \omega)]$$

We simplify an expectation over an infinitely large dataset $\mathcal{D}$ collected by $\pi_b$:

$$\mathbb{E}_{(s_t, a_t) \in \mathcal{D}}[\log \pi_G(a_t \mid s_t, \omega)] = \mathbb{E}_{s_t \in \mathcal{D}}[\lambda \log \pi_G(a_t = a^{(1)} \mid s_t, \omega) + (1 - \lambda) \log \pi_G(a_t = a^{(2)} \mid s_t, \omega)]$$

Since the actions are conditionally independent of the states, let $\hat{p} = \pi_G(a_t = a^{(1)} \mid s_t, \omega)$ for any state $s_t$. Then:

$$\mathbb{E}_{(s_t, a_t) \in \mathcal{D}}[\log \pi_G(a_t \mid s_t, \omega)] = \lambda \log \hat{p} + (1 - \lambda) \log(1 - \hat{p})$$

We can use calculus to maximize the above objective with respect to $\hat{p}$.

$$\frac{d}{d\hat{p}}[\lambda \log \hat{p} + (1 - \lambda) \log(1 - \hat{p})] = \frac{\lambda}{\hat{p}} - \frac{1 - \lambda}{1 - \hat{p}}$$

$$= \frac{\lambda(1 - \hat{p}) - (1 - \lambda)\hat{p}}{\hat{p}(1 - \hat{p})}$$

Setting the derivative to 0:

$$\lambda(1 - \hat{p}) - (1 - \lambda)\hat{p} = \lambda - \lambda\hat{p} - \hat{p} + \lambda\hat{p} = 0 \implies \boxed{\hat{p} = \lambda}$$

This yields an identical policy to the behavioural policy $\pi_b$. Next, consider the derivation of the probability of taking action $a^{(2)}$ conditioned on $\Phi_t$ and $s_t$ based on data $\mathcal{D}$ from $\pi_b$:

$$\Pr_{\pi_b}[a_t = a^{(2)} \mid s_t = s^{(h)}, s_{t+K} = s^{(h+1)}]$$

$$= \frac{\Pr_{\pi_b}[a_t = a^{(2)}, s_{t+K} = s^{(h+1)} \mid s_t = s^{(h)}]}{\Pr_{\pi_b}[s_{t+K} = s^{(h+1)} \mid s_t = s^{(h)}]}$$

In the above step, we used the definition of conditional probability. To compute these probabilities, we recognize that to end up with $s_{t+K} = s^{(h+1)}$ from $s_t = s^{(h)}$, the agent must take action $a^{(2)}$ exactly once between timestep $t$ and $t + K - 1$; any more implies the agent has moved beyond $s^{(h+1)}$ and any less implies the agent is still at $s^{(h)}$.

The probability in the numerator can be written as a product of taking action $a^{(2)}$ at timestep $t$, followed by taking action $a^{(1)}$ at timestep $t + 1$ to $t + K - 1$:

$$\Pr_{\pi_b}[a_t = a^{(2)}, s_{t+K} = s^{(h+1)} \mid s_t = s^{(h)}] = (1 - \lambda) \prod_{t'=t+1}^{t+K-1} \lambda$$

$$= (1 - \lambda)\lambda^{K-1}$$

The probability in the denominator can be written as a product of taking action $a^{(2)}$ at exactly one timestep $t \leq t' < t + K$, followed by taking action $a^{(1)}$ at the remaining timesteps. This can be modeled by a binomial probability where there are $K$ slots to take action $a^{(1)}$, each with probability $1 - \lambda$. Hence:

$$\Pr_{\pi_b}[s_{t+K} = s^{(h+1)} \mid s_t = s^{(h)} = \binom{K}{1}(1 - \lambda)\lambda^{K-1}$$

$$= K(1 - \lambda)\lambda^{K-1}$$

The overall probability is computed as:

$$\frac{\Pr_{\pi_b}[a_t = a^{(2)}, s_{t+K} = s^{(h+1)} \mid s_t = s^{(h)}]}{\Pr_{\pi_b}[s_{t+K} = s^{(h+1)} \mid s_t = s^{(h)}]} = \frac{(1-\lambda)\lambda^{K-1}}{K(1-\lambda)\lambda^{K-1}}$$
$$= \frac{1}{K}$$

We can apply a similar argument to show that $\pi_W^*$ (i.e., at optimum) must clone the derived probability when a maximum likelihood objective is applied. Hence, for the optimal intermediate goal-conditioned policy $\pi_W^*$, we know it obeys:

$$\pi_W^*(a_t = a^{(2)} \mid s_t = s^{(h)}, \Phi_t = s^{(h+1)}) = \frac{1}{K}$$

Since $\pi_G^*(a_t = a^{(2)} \mid s_t = s^{(h)}, \omega) = \pi_b(a_t = a^{(2)} \mid s_t = s^{(h)}) = 1 - \lambda$ and we choose $K < \frac{1}{1-\lambda} \implies \frac{1}{K} > 1 - \lambda$, we conclude that:

$$\pi_W^*(a_t = a^{(2)} \mid s_t, \Phi_t) > \pi_G^*(a_t = a^{(2)} \mid s_t, \omega)$$

This concludes the derivation.

## B  Experimental Details

In this section, we provide more details about the experiments, including hyperparameter configuration, sources of reported results for each method, and details of each environment (i.e., version). For all experiments on WT, the proposed method, we run 5 trials with different random seeds and report the mean and standard deviation across them. On AntMaze and Kitchen, we use goal-conditioning, whereas reward-conditioning is used for Gym-MuJoCo. For all experiments on DT, including Gym-MuJoCo, we run 5 trials with random initializations using the default hyperparameters proposed in Chen et al. [2021] and used in the official GitHub repository. We are unable to reproduce some of the results demonstrated in Chen et al. [2021] and reported in succeeding work such as Kostrikov et al. [2021], Emmons et al. [2021].

### B.1  Environments and Tasks

**AntMaze**  For AntMaze tasks, we include previously reported results for all methods except RvS-G from Kostrikov et al. [2021]. The results for the RvS-G are from Emmons et al. [2021]. We run experiments for DT (reward-conditioned, as per Chen et al. [2021]) and WT across 5 seeds. For all reported results, including WT, AntMaze v2 is used as opposed to AntMaze v0.

**FrankaKitchen**  On Kitchen, we include available reported results from Kostrikov et al. [2021] for all methods except RvS-G and Emmons et al. [2021] for RvS-G, with results omitted for TD3 + BC and Onestep RL as they are not available in other work or provided by the authors. Similarly to AntMaze, we run experiments for DT and WT across 5 seeds. The target goal configuration for WT is "all" (i.e., where all the tasks are solved), per Emmons et al. [2021]. For all reported results, including WT, Kitchen v0 is used.

**Gym-MuJoCo**  On the evaluated locomotion tasks, we use reported results from Kostrikov et al. [2021] for all methods except RvS-R and Emmons et al. [2021] (RvS-R). We run experiments for DT and WT across 5 seeds. The MuJoCo v2 environments are used for all methods.

### B.2  WT Hyperparameters

In Table 3, we show the chosen hyperparameter configuration for WT across all experiments. Consistent with the neural network model in RvS-R/G with 1.1M parameters Emmons et al. [2021], the WT contains 1.1M trainable parameters. For the most part, the chosen hyperparameters align closely with default values in deep learning; for example, we use the ReLU activation function and a learning rate of 0.001 with the Adam optimizer.

In Table 4, we show the chosen hyperparameter configuration for the reward and goal waypoint networks across all experiments. The reward waypoint network always outputs 2 values, the ARTG and CRTG. In general, the goal waypoint network outputs the same dimension as the state since

Table 3: Hyperparameters and configuration details for WT across all experiments.

| Hyperparameter | Value |
|---|---|
| Transformer Layers | 2 |
| Transformer Heads | 16 |
| Dropout Probability (attn) | 0.15 |
| Dropout Probability (resid) | 0.15 |
| Dropout Probability (embd) | 0.0 |
| Non-Linearity | ReLU |
| Learning Rate | 0.001 |
| Gradient Steps | 30,000 |
| Batch Size | 1024 |

it makes $k$-step predictions. Depending on the environment, the goal waypoint outputs either a 2-dimensional location for AntMaze or a 30-dimensional state for Kitchen.

Table 4: Hyperparameters and configuration details for goal and reward waypoint networks across all experiments.

| Hyperparameter | Value |
|---|---|
| Number of Layers | 3 |
| Dropout Probability | 0.0 |
| Non-Linearity | ReLU |
| Learning Rate | 0.001 |
| Gradient Steps | 40,000 |
| Batch Size | 1024 |

## B.3   Evaluation Return Targets

The target return for the Gym-MuJoCo tasks are specified in Table 5, in the form of normalized scores. These were obtained typically by performing exhaustive grid searches over 4-6 candidate target return values, following prior work [Chen et al., 2021, Emmons et al., 2021]. Typically, we choose the range of the grid search based on the interval close to or higher than the state-of-the-art normalized scores on each of the tasks.

Table 5: Normalized score targets for WT on reward-conditioned tasks in Gym-Mujoco.

| Task | Normalized Score Target |
|---|---|
| hopper-medium-replay-v2 | 95 |
| hopper-medium-v2 | 73.3 |
| hopper-medium-expert-v2 | 125 |
| walker2d-medium-replay-v2 | 90 |
| walker2d-medium-v2 | 85 |
| walker2d-medium-expert-v2 | 122.5 |
| halfcheetah-medium-replay-v2 | 45 |
| halfcheetah-medium-v2 | 52.5 |
| halfcheetah-medium-expert-v2 | 105 |

## C   Analysis of Bias and Variance of Reward-Conditioning Variables

We analyze the bias and variance of existing reward-conditioning techniques: a constant average reward-to-go (ARTG) target, as used in Emmons et al. [2021], and a cumulative return target updated with rewards collected during the episode (CRTG), as in Chen et al. [2021]. By analyzing the bias and variance of these techniques, we can determine the potential issues that may explain the performance of methods that condition using these techniques.

Consider the definitions of the true ARTG ($R_a$) and CRTG ($R_c$) below, based on a given trajectory $\tau$ where the length of the trajectory $|\tau| = T$. These definitions are used to train the policy,

$$R_a(\tau, t) = \frac{1}{T-t} \sum_{t'=t}^{T} \gamma^t r_t \tag{3}$$

$$R_c(\tau, t) = \sum_{t'=t}^{T} \gamma^t r_t \tag{4}$$

At evaluation time, it is impossible to calculate $r_{t'}$ for any $t' \geq t$. As a result, we provide ARTG and CRTG targets, $\theta_a$ and $\theta_c$ respectively. At evaluation time, the values of the ARTG and CRTG are estimated and used as follows in Chen et al. [2021] and Emmons et al. [2021]:

$$\hat{R}_a(\tau, t) = \theta_a \tag{5}$$

$$\hat{R}_c(\tau, t) = \theta_c - \sum_{t'=1}^{t} \gamma^t r_t \tag{6}$$

Ideally, the errors of the estimated $\hat{R}_a$ and $\hat{R}_c$ are minimal so as to accurately approximate the true average or cumulative reward-to-go respectively, but that is often infeasible. To characterize the error of each of the evaluation estimates of ARTG and CRTG, consider the decomposition of the error for a particular reward conditioning variable $R$ and estimated evaluation $\hat{R}$ presented in Theorem C.1.

**Theorem C.1.** *The general bias-variance decomposition of the expected squared error between a true $R(\tau, t)$ (e.g., Equations 1 or 2) and an estimator $\hat{R}(\tau, t)$ (e.g., Equations 3 or 4) under the trajectory distribution induced by an arbitrary policy $\pi$ and unknown transition dynamics $p(s_{t+1} \mid s_t, a_t)$ is given by:*

$$\mathbb{E}_\tau[(R(\tau, t) - \hat{R}(\tau, t))^2] = \mathbb{E}[R(\tau, t) - \hat{R}(\tau, t)]^2 + \text{Var}[\hat{R}(\tau, t) - R(\tau, t)]$$

*Proof.* Similarly to the derivation of the standard bias-variance tradeoff, we expand terms and separate into multiple expectations using the linearity of expectation. We leverage the definition of the covariance, $\text{Cov}(X, Y) = \mathbb{E}[XY] - \mathbb{E}[X]\mathbb{E}[Y]$, and variance, $\text{Var}[X] = E[X^2] - E[X]^2$, in several steps.

$$\mathbb{E}_\tau[(R(\tau, t) - \hat{R}(\tau, t))^2] = \mathbb{E}[R(\tau, t)^2] + \mathbb{E}[\hat{R}(\tau, t)^2] - 2\mathbb{E}[\hat{R}(\tau, t)R(\tau, t)]$$

We can simplify the first term using the definition of the variance and the third term using the definition of the covariance.

$$\mathbb{E}_\tau[(R(\tau, t) - \hat{R}(\tau, t))^2] = \text{Var}[R(\tau, t)] + \mathbb{E}[R(\tau, t)]^2 + \mathbb{E}[\hat{R}(\tau, t)^2] - 2\mathbb{E}[\hat{R}(\tau, t)R(\tau, t)]$$
$$= \text{Var}[R(\tau, t)] + \mathbb{E}[R(\tau, t)]^2 + \mathbb{E}[\hat{R}(\tau, t)^2] - 2(\text{Cov}(\hat{R}(\tau, t), R(\tau, t)) +$$
$$\mathbb{E}[R(\tau, t)] \cdot \mathbb{E}[\hat{R}(\tau, t)]))$$

Similarly, we simplify the $\mathbb{E}[\hat{R}(\tau, t)^2]$ term using the definition of the variance and collect terms.

$$\mathbb{E}_\tau[(R(\tau, t) - \hat{R}(\tau, t))^2] = (\mathbb{E}[R(\tau, t)] - \mathbb{E}[\hat{R}(\tau, t)])^2 + \text{Var}[\hat{R}(\tau, t)] - 2\text{Cov}(\hat{R}(\tau, t), R(\tau, t)) +$$
$$\text{Var}[R(\tau, t)]$$
$$= \mathbb{E}[R(\tau, t) - \hat{R}(\tau, t)]^2 + \text{Var}[\hat{R}(\tau, t)] - 2\text{Cov}(\hat{R}(\tau, t), R(\tau, t)) +$$
$$\text{Var}[R(\tau, t)]$$

Equivalently, since $\text{Var}[X - Y] = \text{Var}[X] + \text{Var}[Y] - 2\text{Cov}(X, Y)$, we can rewrite the result as follows.

$$\mathbb{E}_\tau[(R(\tau, t) - \hat{R}(\tau, t))^2] = \mathbb{E}[R(\tau, t) - \hat{R}(\tau, t)]^2 + \text{Var}[\hat{R}(\tau, t) - R(\tau, t)]$$

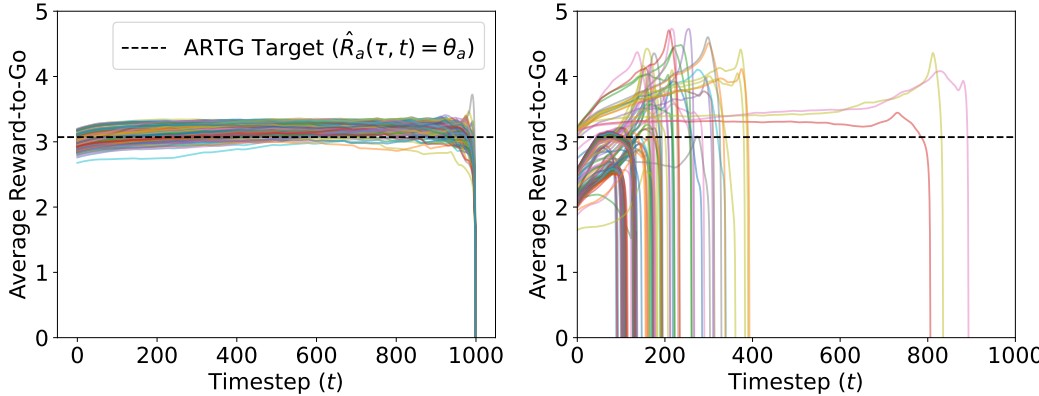

Figure 7: True average reward-to-go as a function of timestep $t$ for 200 rollouts of a transformer policy on `hopper-medium-replay-v2` compared to the constant ARTG target $\theta_a$ (dotted black line), for **left**: successful rollouts and **right**: unsuccessful rollouts.

This completes the derivation of the general bias-variance decomposition. □

**Analysis of ARTG** Consider the decomposition of the error per the bias-variance tradeoff of the ARTG. Trivially, the variance of the estimate in Equation 3 is zero and it is independent of $R$ because it is a constant value, and as a result, the expected error is composed entirely of the bias and irreducible variance.

$$
\begin{aligned}
E_\tau[(R_a(\tau, t) - \hat{R}_a(\tau, t))^2] &= E_\tau[R_a(\tau, t) - \hat{R}_a(\tau, t)]^2 + \text{Var}[R_a(\tau, t)] \\
&= E_\tau[R_a(\tau, t) - \theta_a]^2 + \text{Var}[R_a(\tau, t)]
\end{aligned}
$$

Based on the terms that we are able to minimize, we can derive through calculus that the squared error and bias of a constant estimator are minimized by the mean, i.e., when $\hat{\theta}_a = \mathbb{E}_\tau[R_a(\tau, t)]$. However, in the offline RL setting, we cannot easily determine this value for an arbitrary trained policy $\pi$. As a result, the bias of the technique during evaluation can lead to high error in estimating the true ARTG, which may consequently cause reduced performance during evaluation of the policy.

Empirically, we demonstrate that the high bias of this technique may lead to instability in achieved return across rollouts on `hopper-medium-replay-v2`. Specifically, suppose that a rollout is classified as unsuccessful if it terminates before the time limit $T = 1000$, and analogously, a successful rollout reaches $t = T$ without termination. Based on these distinctions, we display the true ARTG across 200 successful and unsuccessful rollouts of a trained transformer policy in Figure 7.

Clearly, for all successful rollouts, the true ARTG matches the constant target closely across most $t \in [1, T)$, with the exception of $t \approx T$. However, across most failed rollouts, the bias of the constant target for small $t \approx 0$ (i.e., when the policy is taking its first few actions) is relatively high. As $t$ grows, the bias seems to spike upwards significantly and the episode terminates shortly thereafter, indicating an unsuccessful rollout.

Hence, it is evident that when the true ARTG closely follows the prescribed constant ARTG target, the policy consistently achieves state-of-the-art performance. However, whenever the target ARTG underestimates or overestimates the true ARTG by a margin of greater than 0.4, the rollout tends to be unsuccessful, often achieving less than half the return compared to successful rollouts.

To that end, the reward waypoint network uses a neural network formulation to estimate $R_a$ (i.e., without using a constant estimator $\theta_a$) based on the state $s_t$ and the target return $\omega$. By training the neural network to provide a less biased estimate of the ARTG, we show that we can achieve substantial performance improvements over a constant estimate of ARTG on the same task. As shown in Table 6, RvS-R and a constant ARTG both exhibit significantly lower average performance and greater variability compared to WT (with a reward waypoint network).

Table 6: Normalized evaluation scores for different policies and ARTG estimation techniques on the `hopper-medium-replay-v2` task.

| Technique | Normalized Score |
|---|---|
| Transformer (constant ARTG) | $66.5 \pm 15.6$ |
| RvS-R (constant ARTG) | $73.5 \pm 12.8$ |
| WT (waypoint network) | $\mathbf{88.9 \pm 2.4}$ |

**Analysis of CRTG**  Consider a similar decomposition of the error using Theorem C.1 of the CRTG. Though neither the bias nor the variance of $\hat{R}_c$ are necessarily zero, it is important to note that the variance term can be large even if the bias is zero, i.e., if $\theta_c = \mathbb{E}[\sum_{t'=1}^{T} \gamma^t r_t]$.

Corresponding to a best-case scenario where $\hat{R}_c$ is unbiased, we consider a bound of the expected error constructed using only the variance term. For simplicity, we assume that we incur the worst-case variance term with minimal covariance between $R_c$ and $\hat{R}_c$. The resulting dominant quantities are the variance of $\hat{R}_c$ and the irreducible variance of $R_c$.

$$E_\tau[(R_c(\tau,t) - \hat{R}(\tau,t))^2] = \mathbb{E}[R(\tau,t) - \hat{R}(\tau,t)]^2 + \text{Var}[\hat{R}(\tau,t) - R(\tau,t)]$$
$$\geq \text{Var}[\hat{R}(\tau,t) - R(\tau,t)]$$
$$\approx \text{Var}[\sum_{t'=1}^{t} \gamma^{t'} r_{t'}] + \text{Var}[R(\tau,t)]$$

Prior work has shown that Monte Carlo estimates of return exhibit high variance, motivating techniques such as using baseline networks for REINFORCE, $k$-step returns or TD($\lambda$) for TD-learning methods, etc., which aim to reduce variance at the cost of potentially incurring bias Sutton and Barto [1999]. In that vein, we demonstrate that our reward waypoint network predicts a similar quantity as a baseline network and inherits many of its desirable properties.

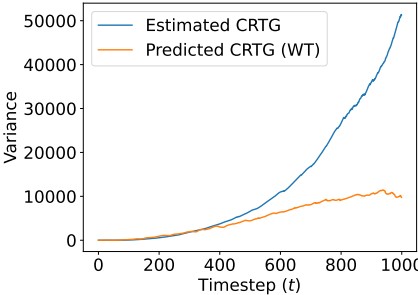

Specifically, while the baseline network predicts $\sum_{t'=t}^{T} \gamma^t r_t$ given $s_t$, we additionally condition our reward waypoint network's prediction on $\omega$. In the offline RL setting where datasets are often a mixture of suboptimal and optimal trajectories, we require that the waypoint network can differentiate between such trajectories. Hence, by conditioning on the overall reward $\omega$, the reward waypoint network is able to differentiate between high and low return-achieving trajectories.

Figure 8: Variance of estimated CRTG (Equation 4) and predicted CRTG by the reward waypoint network on the `hopper-medium-replay-v2`.

Empirically, we demonstrate that the predicted CRTG from our reward waypoint network exhibits lesser variance on `hopper-medium-replay-v2` than the estimated CRTG (i.e., as in Equation 4). As shown in Figure 8, the variance of both techniques are relatively similar until $t \approx 400$, after which the variance of the estimated CRTG appears to grow superlinearly as a function of $t$. In the worst case, at $t \approx T$, the variance is nearly 5x larger than for the predicted CRTG.

# D  Additional Experiments

## D.1  Analysis of Stitching Region Behavior

To add on to the analysis of the goal waypoint network presented in the main text, we analyze the "failure" regions of transformer policies with and without a goal waypoint network. That is, by determining the final locations of the agent, we can examine where the agent ended up instead of the target location. Similar to the analysis in Section 6.3, this analysis can inform the stitching capability of our methods.

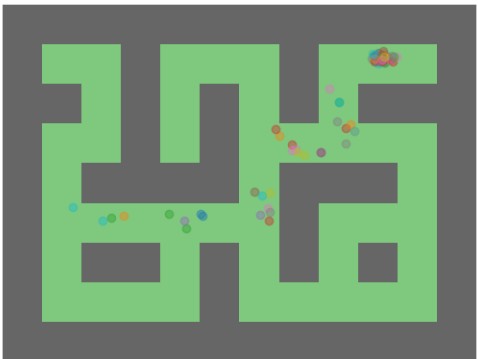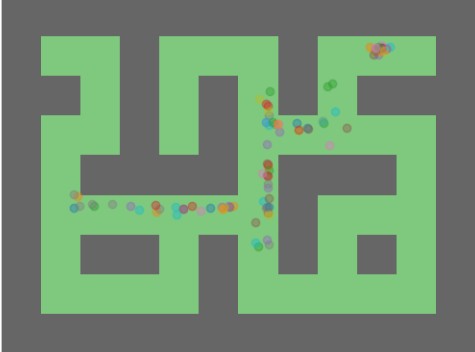

Figure 9: End locations for `antmaze-large-play-v2` during 100 rollouts of **left**: WT and **right**: global goal-conditioned transformer policy.

Based on Figure 9, it is clear that the WT does not get "stuck" (e.g., after taking the wrong turn) as often as the policy conditioned on global-goals. Moreover, the number of ants ending up near the beginning portions of the maze (i.e., the bottom left) is significantly smaller for WT, which contributes to its doubled success rate. We believe these are primarily attributable to the guidance provided by the goal waypoint network through a consistent set of intermediate goals to reach the target location at evaluation time.

Interestingly, we observe that WT displays an increased rate of failure around the final turn relative to other regions in the maze. As there is a relative lack of density in other failure regions closer to the beginning of the maze, we hypothesize that some rollouts may suffer from the ant getting "stuck" at challenging critical points in the maze, as defined in Kumar et al. [2022]. This indicates an interesting direction of exploration for future work and a technique to combat this could result in policies with nearly 100% success rate in completing `antmaze-large-play-v2`.

## D.2 Target Reward Interpolation

Unlike traditional value-based approaches, RvS methods such as DT, RvS, and WT present a simple methodology to condition the policy to achieve a particular target return. In theory, this allows RvS to achieve performance corresponding to the desired performance level, provided that it has non-zero coverage within the offline training dataset.

To examine RvS's capability in this regard, we replicate an analysis performed in Emmons et al. [2021] to examine whether RvS can interpolate between modes in the data to achieve a particular target return. In this case, we compare WT to RvS-R on the `walker2d-medium-expert-v2` task, in which the data is composed of two modes of policy performance (i.e., medium and expert). The mode corresponding to the "medium" data is centered at a normalized score of 80, whereas "expert" performance is located at a normalized score of 110.

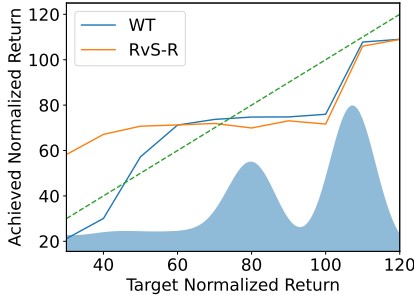

Figure 10: Achieved normalized score versus the target normalized score for `walker2d-medium-expert-v2` task on WT and RvS-R.

Based on Figure 10, WT shows a reasonably improved ability to interpolate in some regions than RvS-R. Where RvS-R displays a failure to interpolate from normalized targets of 30-60 and WT does not, both tend to be unable to interpolate between 70-100 (i.e., between the modes of the dataset).

Although the reasons for the failure in interpolation are unclear across two RvS methods, it is a worthwhile analysis for future work. We hypothesize that techniques such as oversampling may mitigate this issue as this may simply be linked to low frequency in the data distribution.

### D.3 Comparisons to Manual Waypoint Selection

We compare the performance of the proposed goal waypoint network with a finite set of manual waypoints, hand-selected based on prior oracular knowledge about the critical points within the maze for achieving success (i.e., turns, midpoints). Based on the selected manual waypoints, shown in Figure 11, we use a simple algorithm to provide intermediate targets $\Phi_t$ based on a distance-based sorting approach, shown in Algorithm 2.

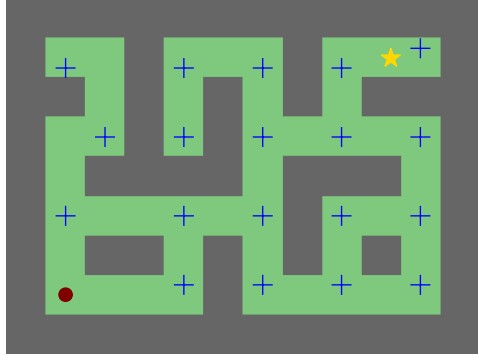

Figure 11: Manually selected waypoints (blue pluses) for `antmaze-large-play-v2`, the chosen task to evaluate the proposed approach. As before, the start location is marked with a maroon dot, and the target location is marked wit a gold star.

---

**Algorithm 2** Manual waypoint selection with $W_m$ and $s_t$ using $L_2$ distance and a given global goal $\omega$.

---

$W_c \leftarrow \{w_m : ||w_m - \omega||_2 \leq ||s_t - \omega||_2\}$ {consider waypoints that brings agent closer to $\omega$}
**return** $\arg\min_{w_c \in W_c} ||w_c - s_t||_2$

---

With all configuration and hyperparameters identical to WT, we compare the performance of a global goal-conditioned policy, WT with manual waypoints, and WT with the goal waypoint network on `antmaze-large-play-v2` in Table 7.

The results demonstrate that WT clearly outperforms manual waypoint selection in succeeding in the AntMaze Large environment. However, while comparing a global-goal conditioned policy and a policy conditioned on manual waypoints, it is clear that the latter improves upon average performance and variability across initialization seeds. We believe that this illustrates that (a) waypoints, whether manual or generated, tend to improve performance of the policy and (b) finer-grained waypoints provide more valuable information for the policy to succeed more.

Table 7: Normalized evaluation scores for different waypoint selection techniques on the `antmaze-large-play-v2` task.

| Technique | Normalized Score |
|---|---|
| No Waypoints | $33.0 \pm 10.3$ |
| Manual Waypoints | $44.5 \pm 2.8$ |
| Waypoint Network | **$72.5 \pm 2.8$** |

We believe that this provides further verification and justification for both the generation of intermediate targets and the procedure of generation through a goal waypoint network that performs $k$-step prediction.

### D.4 Delayed Rewards

An important case to consider for reward-conditioned tasks is when the rewards are delayed (often, provided at the end of the trajectory). By providing nearly no intermediate targets, it is often challenging to complete these tasks. To verify that our design choices to construct modeled intermediate

targets does not reduce performance on configurations in which such information is unavailable, we evaluate WT on 3 MuJoCo tasks where all rewards are provided at the final timestep. Moreover, we ablate the choice of removing actions from WT, which may have an effect on the performance considering that intermediate rewards are not provided.

To compare our performance with other methods, we report results for CQL, DT, and QDT from Yamagata et al. [2022] for all non-expert tasks and from Chen et al. [2021] from `hopper-medium-expert-v2` in Table 8. The results demonstrate that our method is relatively comparable to DT in performance and that removing actions does not significantly affect performance in these cases. All BC methods significantly outperform CQL across the chosen delayed reward tasks.

Table 8: Normalized scores across 5 seeds on delayed reward MuJoCo v2 tasks, where all rewards are provided at the final timestep, on WT with action conditioning ($a_t$) and without action conditioning.

| Environment | CQL | DT | QDT | WT ($a_t$) | WT |
|:---:|:---:|:---:|:---:|:---:|:---:|
| hopper-medium-expert-v2 | 9.0 | **107.3 ± 3.5** | - | 103.9 ± 3.0 | 104.2 ± 2.4 |
| halfcheetah-medium-replay-v2 | 7.8 ± 6.9 | **33.0 ± 4.8** | 32.8 ± 7.8 | 30.6 ± 2.6 | 30.9 ± 2.8 |
| walker2d-medium-v2 | 0.0 ± 0.4 | 69.9 ± 2.0 | 63.7 ± 6.4 | **71.5 ± 0.9** | 70.8 ± 1.7 |

# E    Baseline Reproduction

We reproduce the TD3+BC results reported in Fujimoto and Gu [2021] and subsequently in Kostrikov et al. [2021], Emmons et al. [2021], which are not reproduced by Liu and Abbeel [2023]. For 8 MuJoCo tasks chosen from the same evaluation tasks as WT, we show the normalized score of TD3+BC and the reported mean score as a function of the number of iterations of training. For all of the tasks, it is evident that our reproduction aligns with the original reported result, whereas those in Liu and Abbeel [2023] are consistently higher. We hypothesize that these findings are similar in nature to our inability to reproduce results for DT in 1.

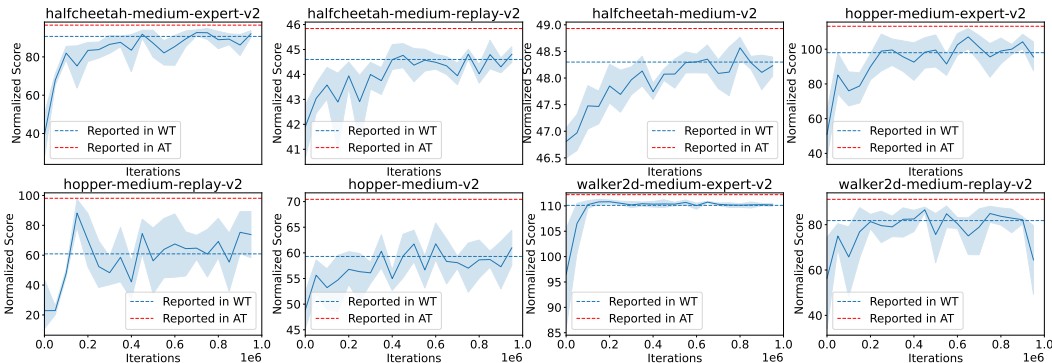

Figure 12: Reproductions of TD3+BC baseline for 8 MuJoCo tasks across 5 seeds for 1M gradient steps, showing that all our reported results (blue, dotted line) lie within the reproduced confidence intervals and most of the results reported in Liu and Abbeel [2023] (red, dotted line) do not.

Moreover, to attempt to reduce the possible effect of hyperparameters on the performance difference, we sweep 16 combinations of the number of actor and critic layers based on the original codebase used by the TD3+BC authors here: https://github.com/sfujim/TD3_BC. Note that their original implementation features a hardcoded 2 layer neural network for actor and critic. In 9, we show that no combination of hyperparameters for the number of actor and critic layers is sufficient to reach the performance reported by Liu and Abbeel [2023] and that as the number of layers increases beyond 3 or decreases beyond 2, the performance decreases on `hopper-medium-replay-v2`.

Table 9: Average normalized score of TD3+BC baseline across 5 seeds by varying number of actor ($L_A$) and critic network hidden layers ($L_C$) on `hopper-medium-replay-v2`.

|        |      | $L_C$ |      |      |
|--------|------|-------|------|------|
| $L_A$  | 1    | 2     | 3    | 4    |
| 1      | 48.7 | **80.1** | 73.1 | 73.9 |
| 2      | 55.2 | 64.4  | 50.2 | 61.3 |
| 3      | 59.8 | 59.1  | 69.6 | 64.6 |
| 4      | 42.9 | 60.5  | 63.6 | 54.4 |

