# OpenReview forum: "Waypoint Transformer: Reinforcement Learning via Supervised Learning with Intermediate Targets"
_NeurIPS.cc/2023/Conference — NeurIPS 2023 poster_

### Official Review · Reviewer_j7or · 2023-07-02

**Soundness:** 2 fair
**Presentation:** 2 fair
**Contribution:** 2 fair
**Rating:** 4
**Confidence:** 4

**Summary:**

This paper analyzes the limitation of the existing decision transformer architecture, i.e., the “stitching” problem associated with global goals and the high bias and variance of reward-to-go targets. To solve this limitation, this paper presents Waypoint Transformer, which predicts the constant average and the cumulative reward-to-go as short-term and intermediate goals, and then guide the transformer to generate trajectories to reach these goals. This method achieves good performance on various benchmarks and significantly reduce training time.

**Strengths:**

> This paper analyzes the limitation of the existing decision transformer architecture – the “stitching” problem, i.e., it cannot stitch segments of suboptimal trajectories into an optimal sequence. To solve this problem, this paper connects the relevant subsequences by providing intermediate targets, which is implemented by predicting the constant and the cumulative reward-to-go.

> This paper achieves SOTA performance in many benchmarks compared with offline RL methods. In addition, it found that we can reduce training time and improve the performance by removing action conditions in original decision transformer.

> The method is easy to be implemented.


**Weaknesses:**

> The novelty of the method is limited. The constant condition, i.e., the average reward-to-go, and the cumulative reward-to-go are widely used in offline reinforcement learning in recent work.

> The presentation of this paper can be improved, e.g., it is difficult to see the average reward-to-go is a constant but the cumulative reward-to-go is updated using rewards obtained during the rollout in Eq. (2). In addition, the relationship between short-term goals and reward-to-go is not clear.

>The authors do not provide a detailed analysis of the failure cases or limitations of their method.

**Questions:**

> WT is not conditioned on past actions. How does WT solve the ‘delayed reward ’case, i.e., an action receives a reward after many interactions?

> Why waypoint network can generate short-term and intermediate goals? How can we ensure that its output can guide the policy towards to ‘stitching region’?

> Why does the waypoint network predict a TD(n) target to achieve the trade-off between a constant ARTG target and the MC estimates for CRTG? Is it a more intuitive and direct way to solve the high bias and the variance?


**Limitations:**

See Weaknesses.

---

> ### Author Rebuttal · Authors · 2023-08-10
>
> We thank the reviewer for their feedback and insightful comments. Below are our responses to your questions.
>
> >The novelty of the method is limited. The constant condition, i.e., the average reward-to-go, and the cumulative reward-to-go are widely used…
>
> We do not dispute that ARTG/CRTG are commonly used (they serve as our baselines), and we cite relevant literature that use these quantities. However, to our knowledge, no work has investigated the issues this presents with regards to bias and variance of the conditioning variable itself and proposed a solution that shows performance competitive with TD learning methods on most tasks. We also believe that conditioning on **predicted** RTGs/goals in offline RL is novel.
>
> While our method is quite intuitive/simple, we provide a new perspective to enable more useful conditioning variables for offline RvS methods and present significant improvements in performance over DT/RvS, on-par or surpassing IQL/CQL/etc.
> >The presentation of this paper can be improved, e.g., it is difficult to see the average reward-to-go is a constant but the cumulative reward-to-go is updated using rewards obtained during the rollout in Eq. (2). In addition, the relationship between short-term goals and reward-to-go is not clear.
>
> We apologize that the presentation is unclear, but we respectfully disagree with your analysis of Eq 2. Eq 2 shows the training objective for the reward waypoint network, where we train against the true concatenated ARTG and CRTG over training trajectories. It does not show that the cumulative RTG is updated since it minimizes a loss. Since the CRTG being updated with rewards increases variance significantly (Sec 4.3, line 192), we do not perform those updates in WT. The ARTG is set to be a constant at evaluation in techniques such as RvS but not in WT (Sec 4.3, line 193).
>
> Short-term goals are defined in the context of goal conditioning (Sec 4.2), where we do not consider rewards. Reward-to-go is only considered in the context of reward conditioning (Sec 4.3). We consider them both as intermediate targets for the paradigms of reward and goal conditioning.
>
> > The authors do not provide a detailed analysis of the failure cases or limitations of their method.
>
> We discuss some of the concrete limitations of the method in Sec 7 based on our results (lines 383-388), but we agree that these are not detailed analyses due to space limitations. There are certainly limitations that continue to exist due to the BC objective, including with distribution shift, multimodality, stochastic environments, and random data. We are happy to add such analyses to the paper.
>
> >WT is not conditioned on past actions. How does WT solve the ‘delayed reward ’case, i.e., an action receives a reward after many interactions?
>
> This is an interesting point, but it is unclear why action conditioning should notably improve performance in this case. Empirically, it seems to have a negligible effect (Table 3 in rebuttal PDF).
>
> Suppose that the rewards are provided at the end of the sequence and we condition on CRTG, which is equal to the return throughout the sequence. The larger the return of the training trajectory, the more likely those actions are to be imitated during evaluation time when we set the target return to be as large.
>
> A transformer policy (conditioned on actions) trains on sequences containing the actions of the behavioural policy. However, during evaluation, the actions placed in the sequence are of the learned transformer policy (not behavioural) which generated those actions, i.e., effectively duplicating info.
>
> If we do not condition on the learned policy’s actions at evaluation time, the hidden states of the transformer should anyway contain information about these past actions (since the transformer policy produced them). Hence, we do not lose much information about the learned policy’s actions.
>
> In Table 3 (rebuttal PDF), we show that including actions has negligible impact across 3 tasks, and we are happy to perform more experiments to further verify.
>
> >Why waypoint network can generate short-term and intermediate goals? How can we ensure that its output can guide the policy towards to ‘stitching region’?
>
> As described in Sec 4.2, the training objective (Eq 1) minimizes error in $K$-step predictions of state $s_{t + K}$ based on the current state $s_t$ and eventual outcome $\omega$ under the behavioural policy. Importantly, since $K$ is chosen to be relatively small compared to the time horizon $H$, the predictions will reflect intermediate states $s_{t+K}$ that should be reached between the current state $s_t$ and the eventual outcome $\omega$. We cannot guarantee it will lead the policy to a stitching region, but it will provide short-term goals that will help provide intermediate guidance on how to achieve the global goal (and that is typically through a stitching region).
>
> >Why does the waypoint network predict a TD(n) target to achieve the trade-off between a constant ARTG target and the MC estimates for CRTG? Is it a more intuitive and direct way to solve the high bias and the variance?
>
> We use a simple BC objective to predict/generate the CRTG and ARTG of trajectories (instead of fixing them to a constant value or updating using rewards). We do not leverage bootstrapping of a value function in any way (i.e., we predict a Monte Carlo TD($\lambda = 1$) target).
>
> As mentioned in Sec 4.3 (lines 204-208), an analogous approach is training a baseline network, which is known to reduce variance with a similarly simple BC objective. We believe this is an intuitive and simple way to reduce variance of PG methods, and we have simply adapted it for usage in offline RL by specifying a target return (since we want to perform conditional BC) and also predicting ARTG (which is scaled by the number of timesteps). To our knowledge, there is not a simpler way proposed to solve this.
>
> Please let us know if you have any more feedback or would like to discuss more!

---

### Official Review · Reviewer_QWFP · 2023-07-05

**Soundness:** 2 fair
**Presentation:** 2 fair
**Contribution:** 3 good
**Rating:** 5
**Confidence:** 3

**Summary:**

This paper proposes Waypoint Transformer for offline reinforcement learning. The novel Waypoint Transformer architecture dynamically generates waypoints for guiding the policy, either in a goal state or a return manner. The authors empirically show that the Waypoint Transformer exhibits better performance over previous reinforcement learning via supervised learning (RvS) algorithms. Meanwhile, numerous ablations are presented in the main text.

**Strengths:**

(a) this paper is easy to read and follow. I appreciate that the idea is quite straightforward, and simple. As far as the reviewer can tell, the presented idea in this paper is novel

(b) it is good to see that sequence modeling algorithms can finally achieve meaningful performance on tasks like antmaze, kitchen. I believe this is the most critical contribution of this work

(c) the authors provide many illustrations to aid the readers understand the claims

(d) the experimental results show that Waypoint Transformer is better than many baseline algorithms

**Weaknesses:**

(a) some claims from the authors are not proper. For example, the authors write that "with performance on par or greater than existing state-of-the-art temporal difference learning-based methods". With the fast advances in the field of offline RL, the chosen baselines in this paper are no longer state-of-the-art. There are many model-free or model-based offline RL algorithms that can achieve far better (often saturated) performance on MuJoCo or other datasets (e.g., ensemble-based algorithms like EDAC).

(b) Table 1 is chaotic, with multiple bolded and highlighted numbers in each row. It is hard to decode whether the waypoint transformer can achieve the best performance on numerous datasets. It seems currently that the performance of waypoint transformer is close to IQL, but I think this is already a big step. Another concern from the reviewer is that the authors do not compare against Q-learning transformer (although cited in the paper). It seems that the authors fits the return-to-go (in Equation 2), and this is similar to fit a value function. I believe this shares some similarities to Q-learning transformer.

(c) thought the authors claim that they use fixed hyperparameters across all of the experiments, they actually use different training paradigms for different datasets (i.e., return conditioned for MuJoCo datasets, and goal conditioned for antmaze and kitchen datasets). This reduces the generality of the waypoint transformer. The reviewer wonders how the waypoint transformer behaves on MuJoCo datasets when conditioned on the goal state.

(d) in ablations, the authors only provide $p_{drop}\in\\{0,0.075,0.15\\}$ and $L\in\\{1,2,3\\}$. I believe this is slightly insufficient to show the sensitivity to the hyperparameters. It is unclear whether waypoint transformer can achieve better performance with larger $p_{drop}$

(e) the minor issues are listed below

 - in Algorithm 1, w.r.t ==> w.r.t.

 - line 119, consider ==> we consider

 - the gradient step of waypoint transformer seems to be much fewer than DT or other algorithms. I believe mroe gradient steps are expected since the performance of many offline RL algorithms are known to collapse with longer gradient steps

**Questions:**

(a) in section 4.1, the horizon is fixed. I think the analysis does not apply when the agent encounter the last few states? For example, if we set $K=5$, and suppose $H=100$, then the conditioned intermediate goal state is invalid if $t\ge 96$, since we cannot get $s_{101}$. Any clarifications here?

(b) in line 223, the authors write that "the WT is not conditioned on past actions $a_{t-k..t}$ (unlike DT)". I actually cannot get the meaning of this sentence. Does the authors mean that DT receive action as input while WT do not? Meanwhile, based on DT paper, it seems DT only receives the $a_t$ as input, while not the past action sequences

(c) based on the ablations in Table 2, it seems it is better not to receive action as input. Any explanation here? Why is it?

(d) how should we choose $K$ in practice, or in real-world applications? How should we determine which to condition, return or goal state?

(e) in appendix B.2, the authors use different gradient steps in Table 1 and 2, and a large batch size 1024. Do these parameters have a large influence on the performance of the waypoint transformer

**Limitations:**

The authors discuss the limitations and future work in the main text

---

> ### Author Rebuttal · Authors · 2023-08-09
>
> We thank the reviewer for their valuable feedback. Below are our responses to your questions.
>
> >There are many model-free or model-based offline RL algorithms that can achieve far better (often saturated) performance… (e.g., ensemble-based algorithms like EDAC).
>
> We should clarify that we are not referring to or competing with ensemble methods or online fine-tuning, and we have added this to the text. That said, we do not dispute that there may be a better state-of-the-art. Based on that, we propose rephrasing to: “performance on par or greater than contemporary/commonly used TD learning methods”. To our knowledge, algorithms like CQL, IQL, TD3+BC, etc. are commonly used due to performance and usability. Our method is simpler to train, as computationally efficient, and has on-par or greater performance than such methods.
>
> > … the authors do not compare against Q-learning transformer.
>
> We have added this comparison to our results. We initially omitted it since QDT has worse performance than a Q-Learning algorithm like CQL (i.e., skipping training the DT entirely) or DT across nearly all tasks. Specifically, QDT only improves upon reference DT’s score in 1/6 tested MuJoCo tasks. On MuJoCo tasks with delayed rewards, QDT demonstrates a higher score on only 1/6 tasks.
>
> > they actually use different training paradigms for different datasets… This reduces the generality of the [WT]. The reviewer wonders how the [WT] behaves on MuJoCo datasets when conditioned on the goal state.
>
> We believe that the option of having goal or reward conditioning increases the generality of WT compared to traditional offline RL methods. Importantly, our method can work even if there are no rewards, unlike many other techniques. If rewards are provided, reward conditioning can be used, and if the task has a specific target state, then goal conditioning can be used.
>
> For MuJoCo, the tasks don’t lend themselves to being formulated as goal-reaching tasks since there is no target (e.g., Kitchen has a target configuration) - Emmons et al. (2021) made the same observation. However, if the tasks were reformulated with a target, we could use goal conditioning with a goal waypoint network. In fact, Ant is an undirected locomotion task that is part of MuJoCo, and AntMaze wraps the ant locomotion task with a clear objective or goal, which allows for goal conditioning.
>
> > It is unclear whether waypoint transformer can achieve better performance with larger p_drop
>
> Thank you for the suggestion. The lower dropout values were chosen largely since it is more standard to have lower dropout values (0.1-0.15) for transformers and we wished to restrict ourselves to more default hyperparameter values whenever possible. We have run ablations on a more broad range of dropout values (Table 1 in rebuttal PDF). For some environments, increasing dropout to be large leads to a significant reduction in performance.
>
> >I believe mroe gradient steps are expected since the performance of many offline RL algorithms are known to collapse with longer gradient steps
>
> Could you clarify? We did not observe any collapses in performance with WT with 100K gradient steps, but the performance was slightly lower (67.3 $\pm$ 5.0 on antmaze-large-play-v2).
>
> > …the conditioned intermediate goal state is invalid if $t \ge 96$, since we cannot get. Any clarifications here?
>
> Apologies - you are absolutely correct. Eqn 1 should be:
> $$
> \arg \min_\phi \sum_{\tau \in \mathcal{D}} \sum_{t = 1}^{|\tau| - K} L_\phi(W_\phi(s_t, \omega), s_{t + K})
> $$
> >Does the authors mean that DT receive action as input while WT do not? … it seems DT only receives the as input, while not the past action sequences
>
> WT does not receive any actions as input. For DT, the causal transformer architecture is able to aggregate information from all past actions since the preceding tokens will all be unmasked during the multi-head attention procedure. That is, it will be able to incorporate information from all states, actions, and rewards preceding that current timestep and the current state.
>
> >based on the ablations in Table 2, it seems it is better not to receive action as input. Any explanation here?
>
> At train time, the actions that will be passed into the transformer will be taken by the behaviour policy. Though we attempt to conditionally mimic the behaviour policy, we will ultimately learn a different policy, i.e., distribution shift. At evaluation time, we instead pass in actions taken by the learned policy, which result in additional distribution shift in the input sequence. We hypothesize this causes performance reduction.
> >how should we choose K in practice… which to condition, return or goal state?
>
> For $K$, our ablations suggest a small fraction of the horizon (e.g., 1-5% of $H$). If rewards are provided, then we can certainly condition on the return. If goals are clearly defined for the task, then it would be valuable to use goal-conditioning. For example, the Gym-MuJoCo tasks don’t permit goal conditioning since there is no “target” per se, but AntMaze has a clear target.
> >…the authors use different gradient steps in Table 1 and 2, and a large batch size 1024. Do these parameters have a large influence on the performance of the waypoint transformer?
>
> The choice of large batch size is based on the findings of Emmons et al. (2021). Our experiments show that the choice of batch size of WT does affect performance but not significantly (512: 66.6 $\pm$ 4.7, 1024: 72.5 $\pm$ 2.8, 2048: 67.5 $\pm$ 5.6 on antmaze-large-play-v2). Training for 100K steps yielded a score of 67.3 $\pm$ 5.0.
>
> As for the different number of training steps for the waypoint network, we chose 40K and 30K gradient steps respectively as they roughly correspond to similar training times. In Sec 6.3, we comment that even 1,000 gradient steps on the waypoint network can achieve a score of ~65 on antmaze-large-play-v2 (line 321).
>
> Please let us know if you have any more feedback or would like to discuss more!

---

> > ### Comment · Reviewer_QWFP · 2023-08-12
> > **responses to the authors**
> >
> > Thanks for your rebuttal. Please find my further comments below.
> >
> > **on the presentation**. Table 1 is chaotic, please make it clearer in the later version.
> >
> > **compare against Q-learning transformer**. The authors reply that "We have added this comparison to our results.", but I cannot find it, where is it?
> >
> > **on the generality of WT**. When commenting "This reduces the generality of the waypoint transformer", the reviewer means that **if both rewards and goals are provided, which one should we choose to condition on? Why should we do so?** The authors can filter the highest return trajectory from the MuJoCo datasets and use this trajectory as goals
> >
> > **ablation study**. Thanks for providing results on different $p\_{drop}$, which is helpful, while how about $L$? How WT behaves if the transformer has more layers?
> >
> > **the gradient step**. The authors can refer to Figure 7 in the EDAC paper [x], where CQL collapses with larger gradient steps. For a fair comparison, I believe reporting results under identical gradient steps is important. These are not expensive experiments.
> >
> > [x] Uncertainty-based offline reinforcement learning with diversified q-ensemble
> >
> > **batch size**. Again, for a fair comparison, the batch size ought to be identical. Based on the recent work on batch size in offline RL [y], a larger batch size may incur better performance.
> >
> > [y] Q-Ensemble for Offline RL: Don't Scale the Ensemble, Scale the Batch Size
> >
> > **On Question (a)**. The authors' response seems to confirm my concern. If Equation (1) is $\arg\min\_\phi \sum\_{\tau}\sum\_{t=1}^{|\tau|-K}L\_\phi(W\_\phi(s\_t,\omega),s\_{t+K})$, what about the last few states? Are they not used for training? Meanwhile, this raises my concern about the validity of Section 4.1, as the probability cannot be calculated then
> >
> > **On Question (c)**. The authors' responses make me quite confused. The authors write "At evaluation time, we instead pass in actions taken by the learned policy, which result in additional distribution shift in the input sequence. We hypothesize this causes performance reduction.", when conditioning on actions, I suppose it means the WT is conditioned on state-action sequences during both training and test. What the reviewer wants to know is **why is it better not to condition on actions? does this apply to other DT-based algorithms? Why is it?** I think the current explanations do not answer these questions.

---

> > > ### Author Response · Authors · 2023-08-13
> > > **Response**
> > >
> > > Thank you for the response.
> > >
> > > > Table 1 is chaotic, please make it clearer
> > >
> > > In our original manuscript, we have removed the blue color and made our optimality condition stricter; we hope this makes the table more readable. Unfortunately, to our knowledge, we cannot update the text.
> > > > compare against Q-learning transformer… I cannot find it, where is it?
> > >
> > > As mentioned, we can unfortunately only provide 1 page containing figures and cannot update the text. On our end, we have added reported results for QDT to Table 1 (Yamagata et al., 2022).
> > > > if both rewards and goals are provided, which one should we choose to condition on? Why should we do so?
> > >
> > > We could simply condition on both rewards and goals by concatenating the outputs of the reward/goal waypoint networks without losing any information.
> > >
> > > If you would like to choose only one, it depends on how strong the reward signal is (e.g., AntMaze rewards are sparse indicators, 1/0, which are not nearly as useful as goal locations). However, we don't see a downside to conditioning on both goals/rewards.
> > > > The authors can filter the highest return trajectory from the MuJoCo datasets and use this trajectory as goals
> > >
> > > Unfortunately, this does not account for the initial state, $s_0$, which may differ at evaluation to the highest return trajectory. For undirected locomotion tasks, where the state may describe the pose/mechanics of a robot, we believe it is unclear what a useful goal should *represent* (there is no "target" per se besides reward). Emmons et al. (2021) contains a similar discussion on this.
> > > > **ablation study**… how about L?
> > >
> > > Apologies, we were unable to include additional results for ablating $L$ in the 1-page PDF/rebuttal due to character limits and have shown results for $L=4,5$ below. More layers do not notably improve results.
> > > | $L$ | hopper-medium-replay | antmaze-large-play | kitchen-mixed  | Average        		 |
> > > |---------------|-------------------------------|-----------------------------|-------------------------|-------------------------|
> > > | 4    		 | **91.1 ± 2.8**  	 | 65.8 ± 3.8     		 | 69.7 ± 1.0 		 | 75.5 ± 2.5 		 |
> > > | 5    		 | 88.8 ± 4.5       		 | 66.7 ± 4.7     		 | 70.0 ± 0.8 		 | 75.2 ± 3.3 		 |
> > > > CQL collapses with larger gradient steps…
> > >
> > > In Fig 7, An et al. (2021) use 1M gradient steps, which is uncommon for transformers since they are far less efficient per gradient step and typically require less gradient steps to train (e.g., DT trains ~1K steps/min, IQL trains 50K steps/min).
> > >
> > > That said, at 400K gradient steps, which is after the CQL collapses occur, we obtain nearly identical results as 100K on antmaze-large-play: 67.2 $\pm$ 3.3. We see no indications of a collapse, but let us know if you’d like to see more extensive experiments on this.
> > > >… the batch size ought to be identical. Based on the recent work on batch size in offline RL [y], a larger batch size may incur better performance.
> > >
> > > It is important to note that transformer batch sizes equal the number of *sequences* (each with $k=20$ states, e.g., batch size of 1024 has $1024*20=20480$ states per batch). The transformer batch sizes we tested in our rebuttal on antmaze-large-play (512, 1K, 2K) roughly match the number of states per batch (s/b) in Fig 7 of [y] (with the addition of one small value).
> > > - 320 s/b ($16*20$): 0.0 $\pm$ 0.0
> > > - 10240 s/b ($512*20$): 66.6 $\pm$ 4.7
> > > - 20480 s/b ($1024*20$): **72.5 $\pm$ 2.8**
> > > - 40960 s/b ($2048*20$): 67.5 $\pm$ 5.6
> > >
> > > > On Question (a)… what about the last few states? Are they not used for training?… this raises my concern about the validity of Section 4.1, as the probability cannot be calculated
> > >
> > > No, those states are not used for training since we do not know the respective future states. Similarly, if we train a dynamics model, we cannot use the last state $s_{|\tau|}$ for training. This is typically not an issue since $K$ is small compared to the time horizon so only 1-3% of the data is unused.
> > >
> > > As for Sec 4.1, since this is an *infinite horizon* task with infinite data, where the goal network deterministically outputs the next state, the probability of reaching $\Phi_t$ conditioned on $\pi_b$ can be perfectly estimated everywhere *except* at the terminal state. To fix this, we could simply pad the end of the sequence with $s^{(H)}$.
> > > > why is it better not to condition on actions? does this apply to other DT-based algorithms?
> > >
> > > We did respond to this in our rebuttal, but we apologize for the confusion. We think adding actions to the sequence causes *additional distribution shift*: that reduces performance. This is because: at *train time*, we condition on actions from the **behavioural** policy; at *test time*, we condition on actions from the **learned** policy. That is, DT will aggregate (using attention) actions that it has likely never seen before since it has never seen its “own” actions during training (only those of behavioural policy). This should apply to all BC algorithms since they all suffer from distribution shift.

---

> > > > ### Comment · Reviewer_QWFP · 2023-08-14
> > > > **responses to the authors**
> > > >
> > > > Thanks for the additional clarifications. Some of the concerns are addressed, please find the further comments below.
> > > >
> > > > > compare against Q-learning transformer
> > > >
> > > > could you present the comparison in your follow-up response?
> > > >
> > > > > on the generality of WT
> > > >
> > > > concatenating the outputs of the reward/goal waypoint networks is fine, but as no empirical results are presented, it remains unclear whether this can best utilize the information (e.g., is it possible that concatenating the outputs of the reward/goal waypoint networks underperforms use only goal/reward information?)
> > > >
> > > > > ablation study
> > > >
> > > > It is much clearer now
> > > >
> > > > > the authors can filter the highest return trajectory from the MuJoCo datasets and use this trajectory as goals
> > > >
> > > > the authors respond that *for undirected locomotion tasks, where the state may describe the pose/mechanics of a robot*, however, since the filtered trajectory is nearly optimal (i.e., high return), the states in the trajectory are the desired poses/positions of a robot. We expect the robot can reach those good states by doing so. The initial state does not seem to be a problem, since we only care what state the agent is expected to reach, and the claimed stitching ability should enable the agent to reach those goals. Moreover, one can simply use $s\_{t+K}$ (e.g., Eqn 1) as goals for mujoco tasks. I expect more explanations from the authors to clarify why this is not a good/reasonable setting.
> > > >
> > > > > the gradient step
> > > >
> > > > If it is possible, I would like to see the results of WT with 1M gradient steps
> > > >
> > > > > on Question (a)
> > > >
> > > > I do not think it is fine to throw the last few data without training them. In some robot tasks (e.g., Adroit domains that this paper does not consider), the robot hand needs to hammer a nail, every step along this trajectory counts.
> > > >
> > > > The authors say that the flaw in section 4.1 can be fixed by augmenting the MDP with some pseudo-states. This is feasible but somewhat weakens the role of section 4.1.
> > > >
> > > > > on Question (c)
> > > >
> > > > Thanks. The explanations seem reasonable. It turns out that it is better not to condition on actions.

---

> > > > > ### Author Response · Authors · 2023-08-16
> > > > > **Response**
> > > > >
> > > > > Thank you for your continued feedback. We hope that we have appropriately addressed your points below. Please let us know if you have any additional feedback or require any clarifications.
> > > > >
> > > > > > could you present the comparison in your follow-up response?
> > > > >
> > > > > Below are the comparisons for all the tasks on which QDT is trained and evaluated. WT performs significantly better in all but one task.
> > > > > |Environment|QDT|WT|
> > > > > |-|-|-|
> > > > > halfcheetah-medium|42.3 ± 0.4|**43.0 ± 0.2**|
> > > > > walker2d-medium|67.1 ± 3.2|**74.8 ± 1.0**|
> > > > > hopper-medium|**66.5 ± 6.3**|63.1 ± 1.4 |
> > > > > hopper-medium-replay|52.1 ± 20.1|**88.9 ± 2.4**|
> > > > > halfcheetah-medium-replay|35.6 ± 0.5|**39.7 ± 0.3**|
> > > > > walker2d-medium-replay|58.2 ± 5.1|**67.9 ± 3.4**|
> > > > >
> > > > > > concatenating the outputs of the reward/goal waypoint networks is fine, but as no empirical results are presented, it remains unclear whether this can best utilize the information…
> > > > >
> > > > > With reward information and a target return of 1, we obtain a score of 72.2 ± 3.9 on antmaze-large-play-v2, which is nearly identical to no reward information. No improvements (or reductions) occur, which is reasonable since AntMaze (as well as Kitchen) has sparse rewards.
> > > > > > …since the filtered trajectory is nearly optimal (i.e., high return), the states in the trajectory are the desired poses/positions of a robot. The initial state does not seem to be a problem, since we only care what state the agent is expected to reach… Moreover, one can simply use $s_{t+K}$ (e.g., Eqn 1) as goals for mujoco tasks.
> > > > >
> > > > > The issue is that the “state” the agent eventually reaches (i.e., a target state) is not necessarily relevant in undirected locomotion tasks where the goal is to maximize reward. For example, such an example of a simple task is to walk as much as possible within a finite horizon by optimally utilizing robotic limbs (e.g., like `walker2d`). The state could describe the mechanics of the robotic limbs. Note that this task is largely periodic since walking involves repetitive motion (i.e., have similar states at every $T’$ timesteps, for some period $T’$).
> > > > >
> > > > > The global “goal” state in this case is not defined since the robotic limbs will have periodic (e.g., repetitive) behaviour over time by taking steps continuously, and the incentive is simply to walk as much as possible, not to achieve some kind of final “limb position” (or to reach some location). We believe it's impossible to fully specify walking maximal distance through any global goal state containing only the robot's mechanics.
> > > > >
> > > > > In terms of intermediate goals (e.g., Eqn 1), they can also be underdetermined in tasks without a clear target (location, observation, etc.) and rewards. For example, if our $K$ matches with the periodicity of the walking task $T'$, our intermediate goal $s_{t+K}$ could simply be similar (or equal) to our current state $s_t$, in which case goal conditioning would not improve upon vanilla BC. Additionally, our formulation of intermediate goals requires a global goal $\omega$, and $\omega$ again seems ill-defined.
> > > > >
> > > > > That said, based on your suggestion, we selected the final observation of the training trajectory with the highest return as the global goal and used Eqn 1 for predicting intermediate goals. The resulting score on `hopper-medium-replay-v2` is 67.2 ± 12.6, which is worse than 10% BC and RvS.
> > > > > > I would like to see the results of WT with 1M gradient steps
> > > > >
> > > > > At 1M steps, the score on the same task is 64.5 ± 4.9. This is slightly worse than 100K/400K but there is no evidence of collapse.
> > > > > > I do not think it is fine to throw the last few data without training them. In some robot tasks (e.g., Adroit domains that this paper does not consider), the robot hand needs to hammer a nail, every step along this trajectory counts.
> > > > >
> > > > > It is important to note that the **policy** is trained on **all** the data. It is also worth noting that the final actions to achieve the target configuration in Kitchen are still important and that we outperform all methods on Kitchen's most challenging configuration.
> > > > >
> > > > > For the waypoint network, it is impossible to determine future states for the final 1-3% of the data since they are near to terminal states. Since short-term goals are not defined for states near enough to the terminal state (e.g., we can't learn/specify a good 20m-away goal when we are 10m away from the finish line), we believe that this is reasonable behaviour.
> > > > > > The authors say that the flaw in section 4.1 can be fixed by augmenting the MDP with some pseudo-states. This is feasible but somewhat weakens the role of section 4.1.
> > > > >
> > > > > The padding is simply a way to represent that we would simply remain at the terminal state for the remainder of time. In any case, note that intermediate goals for improved action selection are unnecessary at the terminal state anyway since there are no actions to select.
> > > > >
> > > > > We do show improved action selection with intermediate goals for non-terminal states, where action selection matters.

---

> > > > > > ### Comment · Reviewer_QWFP · 2023-08-21
> > > > > >
> > > > > > Thanks for your reply. Though my concern about training without the last few data still remains and the presentation of this paper can be further improved, I think this paper has some merits. After careful consideration, I confirm my current rating.
> > > > > >
> > > > > > As an additional note, next time maybe the authors better to provide numerical results in a clear tabular form if the reviewer demands.

---

### Official Review · Reviewer_751f · 2023-07-07

**Soundness:** 2 fair
**Presentation:** 3 good
**Contribution:** 3 good
**Rating:** 4
**Confidence:** 3

**Summary:**

This paper proposed a novel, straightforward approach for generating intermediate waypoints in transformer-based sequence modeling. The waypoint networks are supervised with offline datasets and trained to minimize the mean squared error between the targets and predictions. In addition to the improvements shown in the main results, the authors conducted thorough ablation studies, revealing the correlation between waypoints and RL performance.

**Strengths:**

This paper is well-written and easy to follow. The proposed method is novel and straightforward. The analysis experiment is well-designed and informative.

**Weaknesses:**

My main concern relates to the baseline model's performance.

- Have the authors tuned the baseline model thoroughly? They are much lower than the results reported in AT[1]. For example, the TD3+BC results are different from either AT or the original paper. 10% BC result is also much lower than those reported in AT.
- Ablation results from more tasks is beneficial. The authors only choose one task, so the current results are unconvincing.


[1]: Emergent agentic transformer from chain of hindsight experience. H Liu, P Abbeel - arXiv preprint arXiv:2305.16554, 2023

**Questions:**

- Since the waypoint networks are trained to minimize the MSE, imagining multiple paths equally distributed in the dataset between two states A to B, what kind of waypoint would the network predict at state A?
- Have the author thought about extending the idea to the visual domain?

**Limitations:**

Yes.

---

> ### Author Rebuttal · Authors · 2023-08-09
>
> We thank the reviewer for their feedback and insightful comments. Below are our responses to the reviewer's questions.
>
> > Have the authors tuned the baseline model thoroughly? They are much lower than the results reported in AT[1]. For example, the TD3+BC results are different from either AT or the original paper.
>
> Our TD3+BC results are reported from the v2 results in the original paper (**Table 9** in Fujimoto et al., 2021), and the same results are reported by Kostrikov et al. (2021) and Emmons et al. (2021). Unfortunately, the AT paper does not report most of the hyperparameters they used for TD3+BC (nor the code), and it is unclear whether they use the v0 or v2 versions of D4RL/MuJoCo. It is also worth noting that they only used 3 seeds. We suspect this discrepancy in results could be due to either of the aforementioned reasons.
>
> We are able to replicate all of *our* results for TD3+BC in Table 1 on the v2 environments using the original implementation from Fujimoto et al. (2021) (please see Figure 1 in our rebuttal PDF). Compared to the AT paper’s results, our reproductions are quite different, by around 25-30% in the worst case. To verify that hyperparameters aren’t likely the issue, we search over 16 hyperparameter choices on `hopper-medium-replay-v2` (Table 2 in rebuttal PDF) but are still not close to the AT paper’s reported performance.
>
> Our hypothesis is that the AT paper [1] may have similar evaluation issue(s) as DT since we were similarly unable to reproduce results for DT (Sec 6.1, line 252). Note that DT has similar reported scores in both the AT [1] and DT papers for `hopper-medium-replay-v2` (88.7 and 82.7), yet these are vastly different from our reproduction (43.3) on v2. Others have reported this type of discrepancy in DT as well; please see issue 42 on the official DT GitHub.
> >10% BC result is also much lower than those reported in AT.
>
> Similarly, the hyperparameters/code used for 10% BC are not provided in the AT paper. We verify with the implementation of BC that is used in Kostrikov et al. (2021) and Emmons et al. (2021) and obtain similar numbers as we reported (within 3-4%). We believe this is due to the same reason as before.
>
> > Ablation results from more tasks is beneficial. The authors only choose one task, so the current results are unconvincing.
>
> We are happy to address this. We have expanded our ablations to include `kitchen-mixed-v0` and `hopper-medium-replay-v2`, which represent one task from each of our environment types (Table 1, in rebuttal PDF). We have additionally tested more values of dropout. The results demonstrate that only the lack of dropout, too much dropout (>0.3, whereas 0.1-0.15 is typically standard for a transformer), or having a 1-layer transformer may yield reductions in performance, which is relatively unsurprising.
>
> **Questions**:
>
> > Since the waypoint networks are trained to minimize the MSE, imagining multiple paths equally distributed in the dataset between two states A to B, what kind of waypoint would the network predict at state A?
>
> Given our simple implementation of the goal waypoint network as a feedforward neural network, it would likely take the mean of the future states within both sets of trajectories, which is certainly suboptimal. Unfortunately, since DT (and most BC methods) minimize the MSE in the action space, it would likely suffer from the same kind of issue as well. That is, it would optimize to output the mean of two actions that are equally distributed from one state with the same outcome condition (e.g., return or goal). The waypoint network would not change/fix this inherent issue.
>
> That said, in practise, we have observed that this hasn’t been (empirically) an issue for two potential reasons:
> 1. The chosen $K$ is quite small, which means that the maximum error magnitude due to this averaging/mixing issue can be quite small.
> 2. It is uncommon to have perfectly equally distributed paths. Consider a weighted average of the first set of paths consisting of some proportion $\alpha > 0.5$ and with the second set of paths consisting of $1 - \alpha$ proportion of the dataset. In this case, the optimal value per a BC objective is to weight future states of the first set of paths by $\alpha$. Since $\alpha > 0.5$, the waypoints are likely to drift towards the first set of paths successively more and more.
>
> Finally, as mentioned in Sec 4.2 (line 179), the technique is intended and designed to be quite simple (i.e., easy to setup and train) to provide rough intermediate targets. More complex techniques could mitigate this issue, such as multimodal models using a mixture of Gaussians, for example.
>
> > Have the author thought about extending the idea to the visual domain?
>
> Could you please clarify what you mean by the visual domain? This idea could certainly be applied to Atari or similar environments which have visual input, but the existing DT approach has already shown strong performance in that domain (equals or outperforms CQL for 3/4 games), so our approach is likely less needed. In more recent work, Lee et al. (2022) have demonstrated significantly improved multi-game performance from multi-game DT compared to CQL, BC, and DQN by roughly at least a factor of 2x. We wanted to focus primarily on challenging areas in which the DT lagged behind existing offline RL techniques (AntMaze/Kitchen) or where we could not replicate the official results (MuJoCo v2 tasks).
>
> Please let us know if you have any more feedback or would like to discuss more!
>
> *Lee, Kuang-Huei, et al. "Multi-game decision transformers." Advances in Neural Information Processing Systems 35 (2022): 27921-27936.*

---

> > ### Comment · Reviewer_751f · 2023-08-17
> > **Response**
> >
> > Thank the authors for the detailed response.
> >
> > ### Concerns about the baseline results
> > - While the D4RL v2 datasets offer enhanced metadata as outlined here: https://github.com/Farama-Foundation/D4RL/wiki/Tasks#gym, the difference between v0/v2 shouldn't be substantial.
> > - Though DT results can not be reproduced, I suggest including the DT results from the paper, as these have been recognized and cited in works like Emmons et al. (2021) and Kostrikov et al. (2021).
> >
> > ### Concerns about ablation studies on goal-conditioned tasks
> > - Referring to Figure 5 (right), does an increase in validation loss imply model overfitting on the training data, leading to reduced performance?
> > - In Figure 6 (left), it would be valuable if the authors could present results for additional reward-conditioned tasks.
> >
> > ### Insight into the Waypoint Network's Efficacy (focus on minimizing MSE)
> > - Unlike the proposed method, DT doesn't seem to face the same challenges, given its conditioning context. In the Maze scenario, the suggested waypoint network takes $\(s_t, goal\)$ as inputs and predicts a future state. Due to the imperfect navigation training dataset, this future state can be highly multi-modal. Could the authors provide further analysis on how the waypoint network determines the optimal sub-goal, as Figure 4 (c) depicts? While I understand that most BC methods struggle, particularly with noisy data, shouldn't the approach be to enhance and evolve beyond these challenges rather than introducing another component with similar pitfalls? I would suggest refining the waypoint network to improve its efficiency and overall performance.
> >
> >
> > ### Queries on Method Comparisons:
> > - Considering TT[1] outperforms the introduced methods, what advantages does WT hold over TT?
> > - Would it be feasible to integrate the waypoint network into TT?
> >
> > [1] Michael Janner et al. Offline Reinforcement Learning as One Big Sequence Modeling Problem. Advances in Neural Information Processing Systems, 2021

---

> > > ### Author Response · Authors · 2023-08-18
> > > **Response**
> > >
> > > Thank you for the response. Let us know if you have any further questions.
> > > >the difference between v0/v2 shouldn't be substantial.
> > >
> > > The performance differences between v0/v2 indicate that the differences could be substantial (Fujimoto et al. (2021), Appendix C.3). Comparing v0 and v2 results, from Fujimoto et al. (2021) and Kostrikov et al. (2021) respectively, the performance differs by:
> > > - factor of 3.2x on `walker2d-medium-replay`, 1.7x on `hopper-medium`, 1.9x on `hopper-medium-replay` for TD3+BC
> > > - factor of 4.9 on `walker2d-medium-replay`, 3.4x on `halfcheetah-medium-expert`, 1.3x on `hopper-medium`, 3.3x on `hopper-medium-replay` for CQL
> > > - factor of 6.6x on `walker2d-medium`, 9.0x on `walker2d-medium-expert`, 1.8x on `hopper-medium`/`hopper-medium-expert` for BC
> > >
> > > Note that the implementation of TD3+BC used across the papers are identical (since Kostrikov et al. (2021) obtained their TD3+BC results from Table 9 in Fujimoto et al. (2021)).
> > >
> > > There are also many substantial changes in the data/tasks:
> > > - The average reward across all the training data in tasks with v0 vs. v2 is changed by non-trivial amounts, e.g., the difference is ~20% in `hopper-medium-replay`, `halfcheetah-medium`, and `walker2d-medium`; ~25% in halfcheetah-medium-replay; and ~30% in `walker2d-medium-replay`.
> > > - `hopper-medium-replay-v2` contains double the number of observations/actions in the training data compared to `hopper-medium-replay-v0`.
> > >
> > > >I suggest including the DT results from the paper
> > >
> > > Thank you for the feedback; we can certainly do that.
> > > >Referring to Figure 5 (right), does an increase in validation loss imply model overfitting on the training data, leading to reduced performance?
> > >
> > > No, that is not the case; we did not encounter any overfitting. Apologies if that is unclear. We simply trained the waypoint network for different number of steps to yield different values of the validation error. For example, we trained the waypoint network for 1,000 steps to yield a validation error of ~0.7. By training 40K steps (default), we achieved the lowest validation error of ~0.4.
> > > >In Figure 6 (left), it would be valuable if the authors could present results for additional reward-conditioned tasks.
> > >
> > > We chose `hopper-medium-replay-v2` specifically since it has large performance disparity between TD and BC methods and variability across seeds. Of the remaining tasks, `walker-medium-replay-v2` demonstrates this to an extent. We show the score across 5 seeds for each of the reward conditioning techniques in Fig 6 on this task (we are not allowed to upload figures).
> > > - WT: 67.9 ± 3.4
> > > - ARTG: 54.3 ± 5.7
> > > - CRTG: 61.6 ± 4.0
> > >
> > > Controlling for performance (as in Figure 6, right), the standard deviation of CRTG on `walker-medium-replay` is 1.5x larger than WT's prediction at $t=1000$. Importantly, WT’s variability in the conditioning variable in both this task and `hopper` show a plateau unlike CRTG.
> > > >Could the authors provide further analysis on how the waypoint network determines the optimal sub-goal, as Figure 4 (c) depicts? ...shouldn't the approach be to enhance and evolve beyond these challenges?
> > >
> > > Empirically, we have not observed multimodality leading to poor performance for WT since each action minimally changes the state (e.g., Kitchen mixed/partial have multimodality too) and the policy (and waypoint network) eventually commit to one path as said in our rebuttal; in fact, on such challenging tasks with multimodality, WT performs significantly **better** than other methods. For AntMaze (Fig 4c), the waypoint network chooses areas with more training trajectories (i.e., Fig 2); as in our rebuttal, the waypoints drift towards the path with greater conditional density.
> > >
> > > Besides the lack of necessity in terms of performance, multimodal waypoint networks add a layer of complexity, hyperparameter tuning, and more computational cost. Alongside this, we would want TD learning or multimodal DT for the policy since standard BC policies have the same issues, all of which defeats our motivation/goal of a simple BC framework without much tuning (Sec 1, line 20-25).
> > >
> > > As an ablation, we would be happy to test multimodal WT, but given Fig 4c, we suspect it may not improve results.
> > >
> > > >Considering TT[1] outperforms the introduced methods, what advantages does WT hold over TT?
> > >
> > > TT has significantly higher computational cost (18-36x slower than WT to train), more hyperparameters, and algorithmic components (e.g., autoregressive discretization, beam search, etc.; WT could be implemented in \~20 lines on top of transformer code). Despite that, TT’s average margin of improvement on MuJoCo is minimal and statistically insignificant (WT: 76.9 ± 1.2, TT: 78.9 ± 3.3). They do not report results for AntMaze (without Q-functions) or Kitchen.
> > > >Would it be feasible to integrate the waypoint network into TT?
> > >
> > > TT itself could serve as a waypoint network since it can generate $K$-step predictions using a much more complex transformer architecture; it is already integrated.

---

> > > > ### Comment · Reviewer_751f · 2023-08-21
> > > >
> > > > I appreciate the authors for their comprehensive feedback. I've adjusted my scores accordingly. However, my primary reservation remains with the DT's performance.
> > > >
> > > > #### Would it be feasible to integrate the waypoint network into TT?
> > > > While TT currently operates at the primitive action level, introducing planning at the waypoint level is worth considering. An extended planning horizon could offer enhanced benefits for long-horizon tasks. Additionally, this level of temporal abstraction might be advantageous in improving tasks with dense rewards.

---

> > > > > ### Author Response · Authors · 2023-08-21
> > > > > **Response**
> > > > >
> > > > > Thank you for your help in improving our paper. To address your primary reservation, our reproduced performance for DT aligns more closely with the reproductions in **both** Zheng et al. (2022) and Yamagata et al. (2023), Online DT and Q-Learning DT respectively. Across all tested tasks, our confidence intervals for each task overlap significantly with those of the aforementioned works, whereas they sometimes do not with the original reported results. If it would alleviate your reservations, we are happy to either use reported results from Yamagata et al. (2023)/Zheng et al. (2022) or rerun our evaluation for DT. We will also reach out to the DT authors to see if they've observed such performance degradation across tasks and if there are any solutions.
> > > > >
> > > > > With regards to TT, that sounds like an interesting direction for research in future work. We don't particularly believe that was the focus of our work since TT is computationally expensive as is, and adding more complexity to it would certainly not affirm our intention of presenting a simple BC module for high-performance offline RL. That said, we believe one of our contributions is certainly providing an alternative perspective to conditioning variables in offline RL, which can be used for future work as you've described.

---

### Official Review · Reviewer_8scR · 2023-07-07

**Soundness:** 3 good
**Presentation:** 3 good
**Contribution:** 3 good
**Rating:** 6
**Confidence:** 3

**Summary:**

This work proposes a novel approach for enhancing RL via supervised learning (RVS) by integrating intermediate targets. The main idea is that while standard approaches condition on only the terminal state, the proposed waypoint transformer conditions on automatically generated intermediate waypoints instead, improving spatial compositionality. An additional technique for reducing bias/variance by conditioning on predicted average and cumulative reward to go conditioned on the return and current state is proposed. Experiments show sota performance on many environments, outperforming previous RVS approaches and matching or outperforming other offline RL approaches.

**Strengths:**

* The proposed waypoint approach is novel, simple and effective. To me it is similar to hindsight relabelling, except the goal state is taken as an intermediate state instead of the final state. This makes sense as its more likely the actions taken are optimal with respect to a closer state than a further state. The work shows a good illustrative example to build this intuition as well.
* The proposed proxy reward generation to model both ARTG and CRTG effectively reduce bias/varaince by essentially acting as a baseline network while still enabling performance better than the behavior policy through the return conditioning.
* The experimental results are convincing and showcase improved performance and also stability which is great to see.

**Weaknesses:**

* It is unclear when the spatial vs reward waypoint network is used. It seems like the spatial waypoint can only be applied in specific environemnts and not others. But it's not clear when, and whether both can be used at the same time.
* While intermediate spatial waypoints improve spatial compositionality, more abstract "skill compositionality" seems out of reach with this method.
* I don't understand why the reward waypoint network is called a waypoint network when it doesn't seem like it has anything to do with waypoints. I could potentially see a relationship if the horizon of the reward-to-go was only up to t+K to draw relationship to the intermediate outputs, but what is actually used is the full horizon T as far as I can tell. Cand can the authors be clear how the reward-to-go target is defined and its relation ship to the predicted ARTG and CRTG? Also minor: typo in equation 2, superscript should be t'
* Can the authors compare the behavhior cloning objective of learning the reward-to-go (essentially learning a value function of the behavior policy) compared to alternatives such as Q-Learning transformer? Whle the authors mention they restrict themselves to behavior cloning objectives  (L198), I don't see a strong justification for this.

**Questions:**

Can the authors address the weaknesses and questions above?

**Limitations:**

Yes the authors have adequately addressed limitations.

---

> ### Author Rebuttal · Authors · 2023-08-09
>
> We thank the reviewer for their feedback and insightful comments. Below are our responses to the reviewer's questions.
>
> > It is unclear when the spatial vs reward waypoint network is used. It seems like the spatial waypoint can only be applied in specific environemnts and not others. But it's not clear when, and whether both can be used at the same time.
>
> Apologies for the confusion. We mention in Section 6 (line 238, 242) that the AntMaze/Kitchen tasks use goal conditioning and the MuJoCo tasks use reward conditioning. Hence, we use the goal (“spatial”) waypoint network for AntMaze/Kitchen since that is used for goal-conditioned tasks (Section 4.2, line 150) and the reward waypoint network for MuJoCo since that is used for reward-conditioned tasks (Section 4.3, line 187).
>
> Both could certainly be used at the same time by concatenating the values of the two waypoint networks, but the task would have to be amenable to having reasonable definitions of goals. For example, the MuJoCo tasks (e.g., Hopper) typically do not have any notion of a target state, which makes defining goals for those tasks challenging (Emmons et al., 2021). Similarly, if rewards are provided, then the reward waypoint network can certainly be used.
>
> >While intermediate spatial waypoints improve spatial compositionality, more abstract "skill compositionality" seems out of reach with this method.
>
> This is an interesting observation, but to our knowledge, WT’s behaviour should be largely equivalent to other offline RL techniques in these cases, and depending on how one defines skill compositionality, this could potentially be desirable behaviour.
>
> In the case that different skills are exhibited within the same or different spatial segments, then spatial compositionality should enable composing different skills within the respective spatial segments.
>
> If you are referring to composing different skills unseen in those spatial segments in the dataset, which spatial compositionality would not allow, then that would be out of reach with our method. However, this could actually be a desirable behaviour since it is unclear whether the skill (unseen in a particular spatial segment) would be optimal for that segment since there is no data coverage. As a concrete example, an unobserved obstacle may be that the floor is slippery in regions where the behavioural policy does not employ the skill of zig-zagging; in this case, composing the skill of zig-zagging in slippery areas with other skills could result in suboptimal performance. In a similar vein, offline RL techniques such as CQL penalize out-of-distribution actions (such as executing actions for some “skill” when it is unobserved from particular states) and TD3+BC employs a BC objective for a similar reason.
>
> > I don't understand why the reward waypoint network is called a waypoint network when it doesn't seem like it has anything to do with waypoints. I could potentially see a relationship if the horizon of the reward-to-go was only up to t+K to draw relationship to the intermediate outputs, but what is actually used is the full horizon T as far as I can tell.
>
> Apologies for the confusion - the definition of waypoints here may differ from the conventional  definition of a waypoint. We have defined waypoints as intermediate targets (Sec 4, line 113). In this case, the generated RTGs (even with the full horizon) are intermediate targets (i.e., a target that incorporates intermediate information about the current state, $s_t$).
>
> For reward-conditioned tasks, CRTG can also be considered as providing intermediate targets (or waypoints) since the reward-to-go information is updated at each timestep (unlike ARTG, which is fixed, similarly to a global goal). The critical difference is that our intermediate targets are generated by a model (hence, waypoint network), which demonstrably reduces the variance of the target and improve performance in nearly all cases compared to DT which uses CRTG.
>
> >can the authors be clear how the reward-to-go target is defined and its relation ship to the predicted ARTG and CRTG?
>
> Our proposed reward-to-go target is simply the concatenation of the predicted ARTG and CRTG from the reward waypoint network. As shown in Equation 2, the quantity our reward waypoint network estimates is exactly the concatenated ARTG and CRTG (i.e., we minimize the mean squared error between the concatenated ARTG/CRTG with the output of $W_\phi$).
>
> > Can the authors compare the behavhior cloning objective of learning the reward-to-go (essentially learning a value function of the behavior policy) compared to alternatives such as Q-Learning transformer? Whle the authors mention they restrict themselves to behavior cloning objectives (L198), I don't see a strong justification for this.
>
> The Q-Learning DT (QDT) uses standard TD learning techniques, which unfortunately introduces all the potential training issues associated with these methods (more hyperparameter tuning, training instability, etc. as stated in Sec 1, lines 19-20) and forfeits many of the advantages associated with BC-style algorithms.
>
> Moreover, QDT shows notably worse performance than simply using an off-the-shelf Q-Learning algorithm like CQL (i.e., skipping training the DT entirely) or DT across nearly all environments. Unfortunately, the QDT only improves upon reference DT’s mean score in 1 of 6 tested MuJoCo tasks, with statistically significantly worse performance than DT in 2/6 tasks. Similarly, on MuJoCo tasks with delayed rewards, QDT demonstrates a higher score on only 1/6 tasks.
>
> Hence, we believe that adding Q-Learning to DT complicates the training procedure notably more, without much added value in terms of performance. We add a simpler BC objective whose training is rapid and stable across all tasks without much tuning (Figure 5, line 321) and demonstrate equal or improved performance compared to DT on all tasks.
>
> Please let us know if you have any more feedback or would like to discuss more!

---

> > ### Comment · Reviewer_8scR · 2023-08-13
> > **Response**
> >
> > Thanks to the authors for their detailed response, and many of my concerns are answered. To continue the discussion:
> >
> > **Skill compositionality**:
> > Apologies for not being clear. By "skill compositionality", i was simply referring to stitching trajectories where the states are not only spatial. While spatial compositionality is shown with the spatial network, it's not clear to me that the authors have demonstrated stitching/compositionality for the reward waypoint network. Indeed works like QDT point out that standard reward based DT methods may struggle with stitching without introducing TD/dynamic programming approach. Hence I wondering if the authors could  comment if this limitation still remains with the reward waypoint network.

---

> > > ### Author Response · Authors · 2023-08-13
> > > **Response**
> > >
> > > Thank you for the discussion and for your clarification about skill compositionality. We appreciate all of your feedback and believe it has improved the paper overall. We have addressed your point about non-spatial compositionality below.
> > >
> > > In cases where spatiality is ill-defined (e.g., in reward-conditioned tasks like MuJoCo), it is unclear how we could concretely *visualize* whether compositionality is demonstrated or not. For example, it's difficult to determine *how* WT (or any policy) uses or composes different suboptimal training trajectories in undirected locomotion tasks (e.g., `hopper`).
> > >
> > > However, note that with suboptimal reward-conditioned datasets (e.g., `medium`/`medium-replay`), composing different components of suboptimal trajectories is **required** for good performance since no trajectories by themselves are necessarily optimal. This is why prior BC methods may perform poorly here, whereas CQL, IQL, etc. do not. We demonstrate strong performance on such reward-conditioned tasks with WT, in many cases competing with or outperforming IQL, CQL, etc.
> > >
> > > Hence, we believe this shows WT's compositionality outside of spatial tasks, even if not directly through a visualization. We do not believe it has a limitation in struggling with stitching/compositionality, at least to the extent of other BC methods, based on its similar performance to methods that can stitch.
> > >
> > > We hope that clarified this question. If you have any further questions or concerns, we are more than happy to address them.

---

### Author Rebuttal · Authors · 2023-08-10

We would like to thank all the reviewers again for their valuable insights and feedback. We have carefully considered all comments, addressed each concern to the best of our ability, and believe that the new figures/tables (attached as PDF) and revisions significantly improve the thoroughness of the experiments and clarity of writing.

Based on Reviewer 751f’s remark that some of our baseline methods’ performance do not match with results in Liu et al. (2023), we have reproduced results for these baseline methods, namely TD3+BC (Figure 1, rebuttal PDF) and BC. While we can replicate all of *our* reported results (which match with the original TD3+BC results in Fujimoto et al. (2021), as well as Emmons et al. (2021) and Kostrikov et al. (2021)), we are still unable to achieve the results reported by Liu et al. (2023). We suspect that this discrepancy arises since they used only 3 seeds and may have (partially/completely) used the vastly different, older v0 D4RL/MuJoCo environments. Further, since Liu et al. (2023) do not provide hyperparameters/code for their baselines, we have performed experiments across 16 hyperparameter sets on TD3+BC (Table 2, rebuttal PDF) and verified that no combination yields even close-to-comparable performance to the reported results in Liu et al. (2023).

Moreover, based on feedback from Reviewers 751f, QWFP, and j7or, we have significantly expanded the scope of the ablation studies to include 3 environments from Gym-MuJoCo, AntMaze and Kitchen, with 3 additional tested values for dropout for each environment (Table 1, rebuttal PDF). We have also performed ablations across 3 tasks for the delayed rewards case (Table 3, rebuttal PDF). These expanded ablations reinforce that our initial hyperparameter selections were appropriate.

To clarify points raised by Reviewers j7or and QWFP about the cases in which each of the reward and goal waypoint networks are applicable (e.g., a common query was why goal conditioning was not used for MuJoCo, which is also addressed in past RvS work, Emmons et al. (2021)), we have expanded the descriptions of the use of the waypoint networks in Section 4.2/4.3. In Section 6.3, we have further elaborated on why we hypothesize action conditioning generally does not improve performance. Finally, based on feedback from Reviewers 8scR, j7or, and QWFP, we have extended the results with reported performance from Q-Learning Decision Transformer and enriched our discussion with an in-depth analysis of model limitations.

References: *Liu, Hao, and Pieter Abbeel. "Emergent agentic transformer from chain of hindsight experience." arXiv preprint arXiv:2305.16554 (2023).*

---

### Decision · Program_Chairs · 2023-09-21

**Decision:**

Accept (poster)

**Comment:**

Reviewer opinions are split on this work -- with two recommendations for accept and two for reject. After reviewing the paper and discussion, the AC recommends acceptance. The paper presents a relatively straightforward idea, is well-written, and can achieve meaningful performance in difficult tasks settings where behavior cloning techniques for offline RL have so far faltered.

**Summary of Dissenting Opinions.** One reviewer recommends rejection due to discrepancies in reported performance for the TD3 + BC baseline. The reviewer notes that performance reported in Liu et al. 2023 (Table 2) does not match those presented in the submitted work. Authors responded that the D4RL dataset version was not clearly specified in Liu et al. Consistency with results in Fujimoto et al. (Table 2) and the details in the appendix (C.3) suggest the results in Liu et al. may be on D4RLv0. The experiments in the submitted work use D4RLv2 and are thus comparable to Table 9 from Fujimoto et al., with which they are reasonable close. The AC finds these arguments convincing and does not see this as a reason to recommend rejection. The other reviewer recommending rejection did not participate in the author discussion or update their review. Some concerns expressed were requests for clarification or improved presentation. To the AC's judgement, the authors response seems to satisfy their stated concerns.